# Spatial periodicity in grid cell firing is explained by a neural sequence code of 2-D trajectories

**Rebecca RG[1], Giorgio A Ascoli[2], Nate M Sutton[2], Holger Dannenberg[2]\***

[1]Department of Mathematical Sciences, George Mason University, Fairfax, United States; [2]Department of Bioengineering, George Mason University, Fairfax, United States

**\*For correspondence:**
hdannenb@gmu.edu

**Competing interest:** The authors declare that no competing interests exist.

## eLife Assessment

This **valuable** study presents a theoretical framework in which spatial periodicity in grid cell firing emerges as the optimal solution for encoding two-dimensional spatial trajectories via sequential neural activation. The idea is supported by **solid** evidence, though it rests on several key assumptions that merit careful consideration. This work will be of interest to neuroscientists investigating the neural mechanisms underlying spatial navigation.

**Abstract** Spatial periodicity in grid cell firing has been interpreted as a neural metric for space providing animals with a coordinate system in navigating physical and mental spaces. However, the specific computational problem being solved by grid cells has remained elusive. Here, we provide mathematical proof that spatial periodicity in grid cell firing is the only possible solution to a neural sequence code of 2-D trajectories and that the hexagonal firing pattern of grid cells is the most parsimonious solution to such a sequence code. We thereby provide a likely teleological cause for the existence of grid cells and reveal the underlying nature of the global geometric organization in grid maps as a direct consequence of a simple local sequence code. A sequence code by grid cells provides intuitive explanations for many previously puzzling experimental observations and may transform our thinking about grid cells.

## Introduction
### What is the nature of the problem being solved by grid cells?

Grid cells in the medial entorhinal cortex (MEC) (*Fyhn et al., 2004*; *Hafting et al., 2005*) and adjacent regions such as the pre- and parasubiculum (*Boccara et al., 2010*) are hypothesized to supply a 'spatial metric' for cognitive map-based navigation and path integration (*Fuhs and Touretzky, 2006*; *McNaughton et al., 2006*; *Moser et al., 2017*), episodic memory (*Hasselmo, 2009*; *Eichenbaum and Cohen, 2014*; *Schiller et al., 2015*), and navigating abstract feature spaces (*Constantinescu et al., 2016*; *Hawkins et al., 2018*; *Rueckemann et al., 2021*). Grid cells recorded in freely foraging animals fire at multiple locations in space so that the firing fields form a hexagonal lattice (*Figure 1A–D*). Since their first discovery, the remarkable spatial periodicity in grid cell firing has been a topic of intense research (*Moser et al., 2017*). This research was fueled in part by awe of the beautiful symmetry and complexity displayed in the spatial firing pattern of a single cell. Moreover, research efforts were driven by the hope that understanding the underlying nature of grid cell firing would greatly advance our understanding of how the mammalian brain performs navigational computations and higher cognitive functions such as episodic memory.

However, no theory exists to date that explains the emergence of grid cell firing patterns from basic principles, unifying the many seemingly unconnected experimental observations on periodic firing patterns and their distortions in a single framework (*Jeffery, 2024*; *Ginosar et al., 2023*). Consequently, currently existing mechanistic models of grid cell firing fall short of explaining the basic question of *why* grid cells exist in the first place. Most previous studies on grid cell function follow a traditional approach based on the well-founded assertion in biology that structure determines function. This approach has resulted in a vast literature describing properties of grid cells in multiple species, how these properties depend on internal neural circuit dynamics and external cues, and how grid cell dynamics can be employed for computational or behavioral functions such as a population code for spatial location (*Sreenivasan and Fiete, 2011*; *Mathis et al., 2012*; *Wei et al., 2015*), memory-guided navigation (*Fuhs and Touretzky, 2006*; *McNaughton et al., 2006*; *Moser et al., 2017*) and the planning of direct trajectories to goals (*Bush et al., 2015*; *Erdem and Hasselmo, 2014*). However, this traditional approach has been proven notoriously difficult in identifying a teleological cause for the existence of grid cells in mammalian brains, that is, an explanation of the specific functional purpose they serve. Computational models of grid cells such as oscillatory interference models (*O'Keefe and Burgess, 2005*; *Burgess et al., 2007*; *Giocomo et al., 2007*; *Hasselmo et al., 2007*; *Burgess, 2008*) and continuous attractor models (*Fuhs and Touretzky, 2006*; *Burak and Fiete, 2009*; *Shipston-Sharman et al., 2016*) provide mechanistic explanations of how the spatial periodicity in firing can emerge from the structure of microcircuits within the superficial layers of the MEC (*Zilli, 2012*; *Couey et al., 2013*; *Zutshi et al., 2018*). However, these mechanistic models, too, cannot identify the teleological cause for the emergence of spatial periodicity in grid cell firing. Consequently, functions of grid cells are often explained in generic terms and statements, such as that grid cells supply a path integration-based 'metric for space' or provide a 'coordinate system' for spatial mapping. The influential hypothesis that grid cells provide a universal map for space is challenged by experimental data suggesting a yet-to-be-identified local computational function of grid cells (*Jeffery, 2024*; *Ginosar et al., 2023*). Here, we identify this local computational function as a trajectory code.

The approach taken in this study in developing a theoretical framework for grid cells differs from mechanistic modeling approaches and from traditional approaches in the grid cell literature that aim to assign functions to grid cells based on their properties. Instead, we turn the question on its head and ask the reverse question: What is an important function performed by the mammalian brain that could either not be performed at all or would be substantially more costly or inefficient to perform in the absence of grid cells? This approach follows the logic proposed by David Marr: "To phrase the matter in another way, an algorithm is likely to be understood more readily by understanding the nature of the problem being solved than by examining the mechanism (and hardware) in which it is embodied" (*Marr, 2010*, Chapter 1.2).

This study provides mathematical proof that, in Marr's words, "the nature of the problem being solved" by grid cells is coding of trajectories in 2-D space using cell sequences. By doing so, we offer a specific answer to the question of why grid cell firing patterns are observed in the mammalian brain. Thus, we (1) provide a teleological cause for the existence of grid cells and (2) provide a unifying theoretical framework that spans a bridge between the literature on grid cells and the vast literature on neural sequences in the hippocampal formation.

## Results

### Spatial periodicity in grid cell firing emerges from a cell sequence code of trajectories in 2-D space

To constrain our search for a function that requires grid cells or could not be performed efficiently without grid cells, we reasoned that this function shall rely on sequential activation of grid cell assemblies, in short grid cell sequences. This reasoning rests on well-established experimental data on neural activity in the hippocampal formation showing that cell sequences can provide a code for transitional structures of world states, generating an internal representation for memory-guided navigation (*Liu et al., 2023*; *Buzsáki, 2010*; *Buzsáki et al., 2022*). Please note that the term 'cell sequences' refers either to sequences of single cells or the sequences of cell assemblies, which are often observed in animal experiments as sequential activity of individual cells due to limitations in the number of

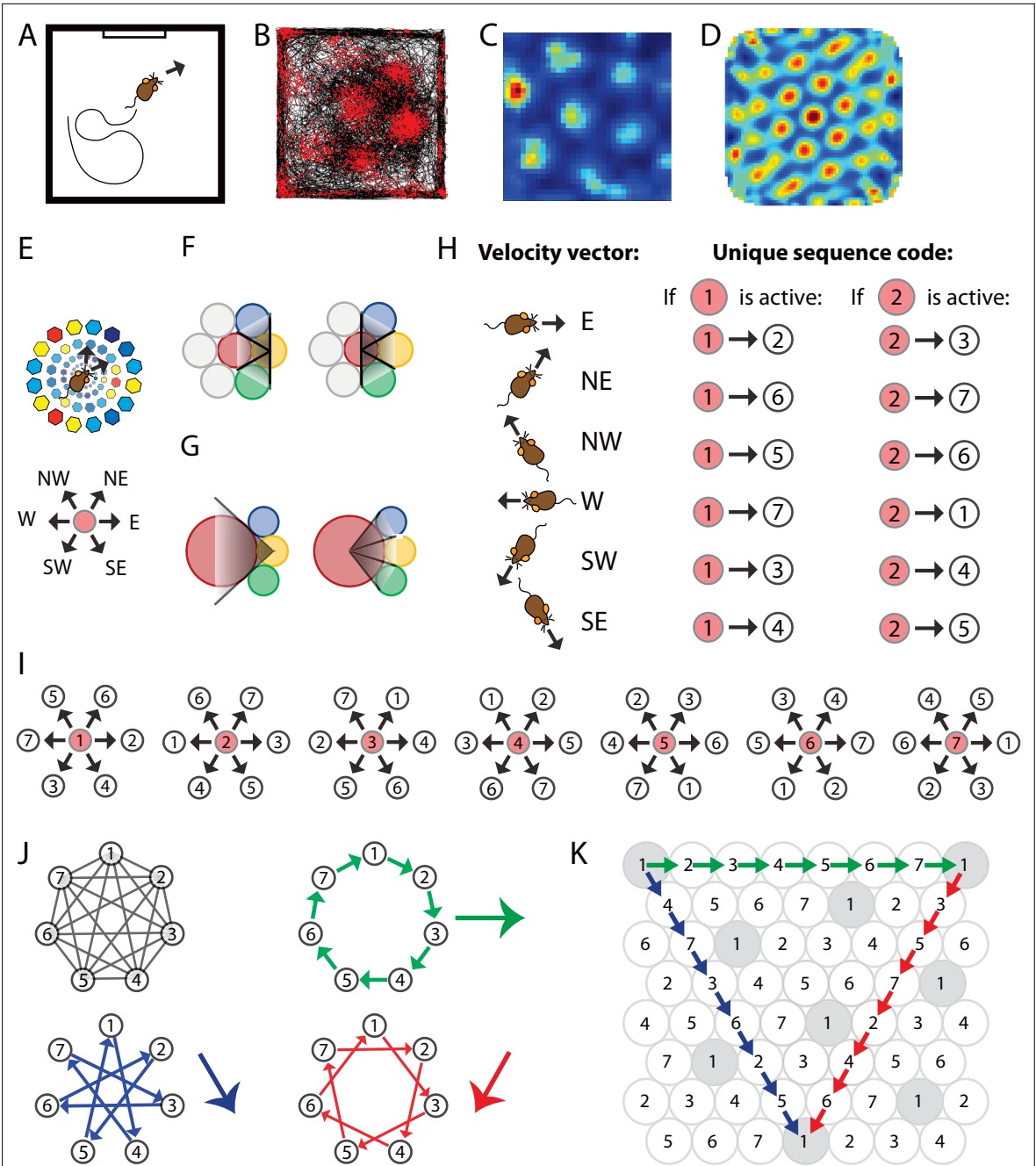

**Figure 1.** Spatially periodic firing of grid cells in 2-D space emerges from a neural sequence code of trajectories. (**A**) Schematic drawing of a 1 × 1 m² environment surrounded by walls in the presence of a single visual cue card. (**B**) Trajectory plot visualizing grid cell spiking activity as a function of the animal's location in space. Data obtained from the medial entorhinal cortex of a freely foraging mouse (*Dannenberg et al., 2020*; *Dannenberg et al., 2019*). The black line indicates the path taken by the animal. Red dots indicate the locations where action potentials (spikes) were generated by the grid cell. (**C**) Firing rate map of the spiking data shown in (**A**). Data are visualized as 3 × 3 cm² spatial bins, smoothed with a Gaussian kernel. Red and blue colors indicate high and low firing rates. Peak and average firing rates are 15 and 1.9 Hz. (**D**) Spatial autocorrelogram of the data shown in (**B**). Red and blue colors indicate high and low correlation values. (**E**) The animal's current position and velocity determine the animal's trajectory. In a sequence code, directions are represented by pairs of sequentially active cell assemblies with non-overlapping cellular composition. We refer to the individual members in a set of completely distinct cell assemblies as *elements*. The red circle represents the currently active element. The next active element in the sequence code is *uniquely* determined by the currently active cell and the velocity vector (Definition 8). N = north, NE = northeast, NW = northwest, W = west, SW = southwest, and SE = southeast. (**F**) Elements' firing fields that surround each other symmetrically in a hexagonal lattice packing allow for equal angular resolution in the coding of trajectories by cell sequences. Colors represent distinct elements. (**G**) If an element's firing field was surrounded by fewer or more than six firing fields that touched each other, opposite directions would not be represented at the same angular resolution.

*Figure 1 continued on next page*

*Figure 1 continued*

In this example, *east → west* cannot be distinguished from *east → northwest* and *east → southwest* because the three directions are represented by the same sequence of elements, *yellow → red*. However, the opposite direction, *west → east*, is represented by three different cell sequences allowing for a finer angular resolution in the representation of traveling direction. (**H**) Example of possible pairs of elements creating a sequence code. If the sequence code shall represent all directions with equal angular resolution, the cellular composition of sequentially active cell assemblies needs to be completely distinct every 60° (**E, F**). Possible representations of velocity vectors by sequences of two elements are shown for when the currently active element is #1 or #2. Numbers within circles identify distinct elements. (**I**) Example of a complete sequence code that uniquely represents all directions at 60° resolution as a function of the currently active element and the traveling direction. (**J**) Sequential activation of seven distinct elements can code for infinitely long trajectories along all three major axes of a hexagonal lattice. (**K**) The same sequential activation map as in (**I**) and (**J**) but plotted in 2-D space. Note that the firing fields of individual elements form grid maps in 2-D space, as highlighted in gray for element #1.

The online version of this article includes the following figure supplement(s) for figure 1:

**Figure supplement 1.** Trajectory coding by cell sequences requires a dense packing of convex firing fields.

neurons that can be recorded simultaneously. We further reasoned that, if a fundamental function of grid cells exists, such a function is very likely related to coding of trajectories. This reasoning rests on theoretical work demonstrating that theta sequences of grid cell populations can provide a traveling-direction signal (*Zutshi et al., 2017*), and that temporally structured neural activity in the hippocampal formation may be necessary to temporally bind neural representations of contiguous events (*Rueckemann et al., 2021*). Moreover, data obtained from rodent experiments (*Allen et al., 2014*; *Gil et al., 2018*) and theoretical work (*Sorscher et al., 2023*) suggest that grid cells serve path integration and memory-guided navigation. A code for spatial trajectories by cell sequences that is usable across different environments would be very useful to keep track of changes in location relative to a starting point, that is, path integration.

We therefore set out to provide mathematical proof that spatial periodicity in grid cell firing emerges from the assertion that cell sequences code for trajectories in 2-D space under the constraint that the number of neurons in the brain is finite.

## Mathematical proof

In the following, we prove that the firing fields of grid cells emerge from assuming a basic set of axioms defining how cell sequences code for trajectories in 2-D space. In particular, we prove that the grid fields must be arranged such that their centers are placed at the points of a hexagonal lattice. We will draw on geometry and combinatorics for this, enabling us to give a visual but nevertheless rigorous argument.

### Definition 1.1: Spatial firing field

A spatial firing field of a neuron defines a specific region in space, where a neuron is active, that is, the neuron responds to a specific spatial location of an animal with an increased rate of action potential firing.

### Remark 1.2

Conversely, if the firing fields of multiple neurons overlap at one point in space, multiple neurons are active at the same time.

### Definition 1.3: Cell assembly

We refer to the co-activity of multiple neurons with overlapping firing fields as a functional *cell assembly*.

### Definition 1.4: Compound spatial firing field

A compound spatial firing field is the union of all firing fields of all cells that are part of a cell assembly.

### Definition 2.1: Isometry

An *isometry* is a bijective map of the plane $f$ that preserves distance, that is, for any two points $x$ and $y$,

$$d\left(f\left(x\right),f\left(y\right)\right)=d\left(x,y\right).$$

We say that two geometric figures (i.e., shapes in the plane) are *geometrically congruent* if there is an isometry of the plane that sends one figure to the other. We will abbreviate this to *congruent* in what follows.

It is well known that two figures are congruent if one can be laid on top of the other so that they match perfectly. This can be done via rotation, translation, reflection, or a combination of these.

### Definition 3.1: Kissing number

Given a geometric figure, the kissing number is the largest number of non-overlapping figures congruent to the original geometric figure that can be arranged in the plane so that they all touch the original geometric figure.

### Remark 3.2

Kissing numbers are usually computed for spheres embedded in various Euclidean spaces, but here we restrict to the plane and allow convex (see Definition 5) geometric figures other than circles.

### Example 3.3

The kissing number of a circle in the plane is 6, which is achieved using a hexagonal packing of circles (*Thue, 1910*; *Fejes, 1942*).

### Definition 4.1: Lattice and lattice packing

A *lattice* in the plane is an infinite set of points such that adding or subtracting any one point to/from another in the lattice returns another point in the lattice, any two points of the lattice are separated by a minimum distance, and any point in the plane is within a maximum distance of a lattice point.

A *lattice packing* is a packing of the plane by congruent geometric objects, such that the centers of the objects are located at the points of a lattice.

### Example 4.2

The points in the plane with integer coordinates form a lattice.

### Example 4.3

The tiling of the plane by equilateral triangles, squares, or hexagons is a lattice packing. So is the hexagonal circle packing, in which every circle is surrounded by six other circles touching it.

### Fact 4.4

A lattice is invariant under isometries that take one point of the lattice to another point of the lattice.

### Fact 4.5

There are five types of a lattice in the plane (*Kittel, 1966*, Chapter 1) (*Figure 1—figure supplement 1A–E*).

1. Oblique
2. Square
3. Hexagonal
4. Rectangular
5. Centered rectangular

### Definition 5: Convexity

A geometric object in the plane is *convex* if for any line segment whose endpoints lie on the object, the entirety of the line segment also lies on the object (*Figure 1—figure supplement 1E and F*).

### Definition 6: Trajectory

A *trajectory* is a path taken in space.

### Remark 6.1

Note that a code for trajectories is independent of the animal's spatial location, consistent with the definition of path integration. This implies that, if the number of neurons is finite (Axiom #4) and the space is large, sequences must eventually repeat in different location, resulting in neural sequences coding for the same trajectories at different locations.

### Definition 7.1: Cell sequence

A *cell sequence* refers to continuous neural activity consisting of sequentially active cells or cell assemblies.

### Definition 7.2: Elements within a cell sequence

In the context of a cell sequence, an 'element' within a sequence refers either to a single cell or a cell assembly. A sequence of cell assemblies refers to the sequential activity of cell assemblies such that the cellular compositions of two sequentially active cell assemblies do not overlap. Conversely, the firing fields of all cellular components of the first cell assembly do not overlap with the firing fields of all cellular components of the second cell assembly.

### Remark 7.3

Cell assemblies can perform pattern completion and are thereby resistant to noise so that a cell assembly would be activated robustly even if a single cell of this assembly failed to be activated. Moreover, cell assemblies allow for conjunctive coding by individual cells.

### Definition 8: Sequence coding of trajectories

*Sequence coding of trajectories* is given if the identity of the next active element in a sequence is uniquely determined by the currently active element and the velocity vector associated with the current trajectory (*Figure 1E–J*).

### Axioms (Table 1)

Our goal is to provide a mathematical proof that spatial periodicity in grid cell firing emerges as a parsimonious solution to provide a code for trajectories in 2-D space by cell sequences.

In the following sections, we demonstrate that if we assume axioms 1–4, it must be the case that the intersecton of all firing fields of trajectory-coding cells or all compound firing fields of cell assemblies must be congruent (the same shape), and the centers of the firing fields must be arranged in a hexagonal lattice. For the sake of simplicity in writing, we refer to cells or cell assemblies as elements.

To illustrate this, note that by Axiom 3, a particular firing field U is surrounded by the maximal number of other firing fields, each of which must be congruent to each other. This implies that all firing fields around U are congruent. Pick a firing field V adjacent to U. By the same argument, it must be the case that all firing fields around V are congruent. Since U is adjacent to V, it is congruent to the firing fields around V. Some of these are adjacent to U as well, so U is congruent to the firing fields around

**Table 1.** Axioms from which grid cell firing emerges as the most parsimonious solution to provide a code for trajectories in 2-D space.

| | Axiom | Remark/example |
|---|---|---|
| Axiom 1 | Cell sequences code for trajectories in 2-D space | This axiom states that the sequential activity of two cells, e.g., $i \to j$, can unambiguously be interpreted by a downstream reader mechanism as a code for one and only one trajectory, e.g., moving from place A to place B |
| Axiom 2 | The reverse cell sequence codes for the reverse trajectory | If the cell sequence $i \to j$ codes for moving from place A to B, then the reverse cell sequence $j \to i$ codes for moving from place B to place A |
| Axiom 3 | Each cell's firing field is surrounded symmetrically by other cell's firing fields so that the angular resolution is constant and maximal across all directions | This axiom means that a sequence code should represent each direction equally with no 'gaps' in the representation (*Figure 3D and E*) |
| Axiom 4 | The number of cells is finite | Grid cells are densest in layer II of the medial entorhinal cortex (*Sargolini et al., 2006*), and this layer has been estimated to contain 24,000 and 58,000 neurons in mice and rats, respectively (*Gatome et al., 2010*) |

both V and U, as well as to V. By continuing this argument across the whole plane, we see that all firing fields must be congruent.

Since each firing field U must be surrounded by the maximal number of firing fields, the kissing number of the arrangement must be maximal. Further, by the symmetry requirement (Axiom 3), this must be achieved by a hexagonal lattice packing with a kissing number of 6 (*Figure 1F and G*). Any square lattice packing either has a kissing number of 4 or lacks symmetry between the 'corner' firing fields and 'side' firing fields 25A1.

We now use Axiom 1 to demonstrate that the firing fields of trajectory-coding elements must be convex. Examining *Figure 1—figure supplement 1H*, we see that if a firing field is non-convex, a single sequence will code for more than one trajectory.

At this point, we know that (i) all firing fields are congruent to each other, (ii) they are arranged in a hexagonal lattice, and (iii) each firing field is convex.

## Definition 9: Row and diagonal

We refer to a straight sequence of firing fields along one of the major axes of the hexagonal lattice as a row and along the remaining two axes as diagonals (*Figure 1K*).

As we will prove below, the possibilities of arrangements of firing fields of all distinct elements are restricted by axioms 1–3. We further realize that the elements appearing in one row of the lattice must repeat if the number of elements is finite (Axiom 4).

But as soon as one element repeats, the *sequence* of elements must also repeat. For example, assume we start in a firing field of one trajectory-coding element and travel in some direction. If, for example, elements 1, 3, and 4 fire in that order and the animal continues traveling in the same direction, elements 1, 3, and 4 will eventually fire again in that order.

Generally speaking, starting in a firing field of element $i$ and going along any set of firing fields, some element must eventually become active again since the total number of elements is finite by Axiom 4. Once there is a repeat of one element's firing field, the whole sequence of firing fields of all elements must repeat by Axiom 1. More specifically, if we had a sequence $1, 2, \ldots, k, 1, t$ of elements, then $1, 2$, and $1, t$ both would code for traveling in the same direction from element 1, contradicting Axiom 1.

The repeating sequence of locally active elements must at least have length 3 as if we had a repeat of length 2, we would have a sequence $i, j, i$ along one trajectory, where $i, j$ would represent two distinct trajectories, contradicting Axiom 1. If the same element fired twice in a row, say we had a sequence $i, 1, 1, j$, then $1, 1$ and $1, j$ would both code for the same trajectory, contradicting Axiom 1 again 25A1.

It follows that the arrangement in one row determines the possibilities of arrangements in the remaining rows.

More explicitly, assuming axioms 1 and 4, the firing fields of trajectory-coding elements must be spatially periodic, in the sense that starting at any point and continuing in a single direction, the initial sequence of locally active elements must eventually repeat with a repeat length of at least 3 (*Figure 2*).

## Remark 11.3

This does not require symmetry in the sense that traveling in two different directions from element $i$ may give repeating sequences of two different lengths, as seen in *Figure 2B*, where the repeats have lengths 4 and 8, depending on the angle of travel from a given element.

The next goal is to determine how our axioms restrict the arrangement of firing fields in a single computational element of trajectory-coding elements. Since these firing fields are arranged in a hexagonal lattice, we determine the possible ways to label the firing fields in this lattice (drawn as circles for convenience), as visually represented in the figures.

Many of the proofs in the rest of this section use the method of proof by cases: we describe all possible ways to label the firing fields, and then explain why some of those cases contradict our hypotheses. A key observation is that the sequence that appears in one row of the lattice restricts the possibilities for which sequences can appear in the adjacent rows.

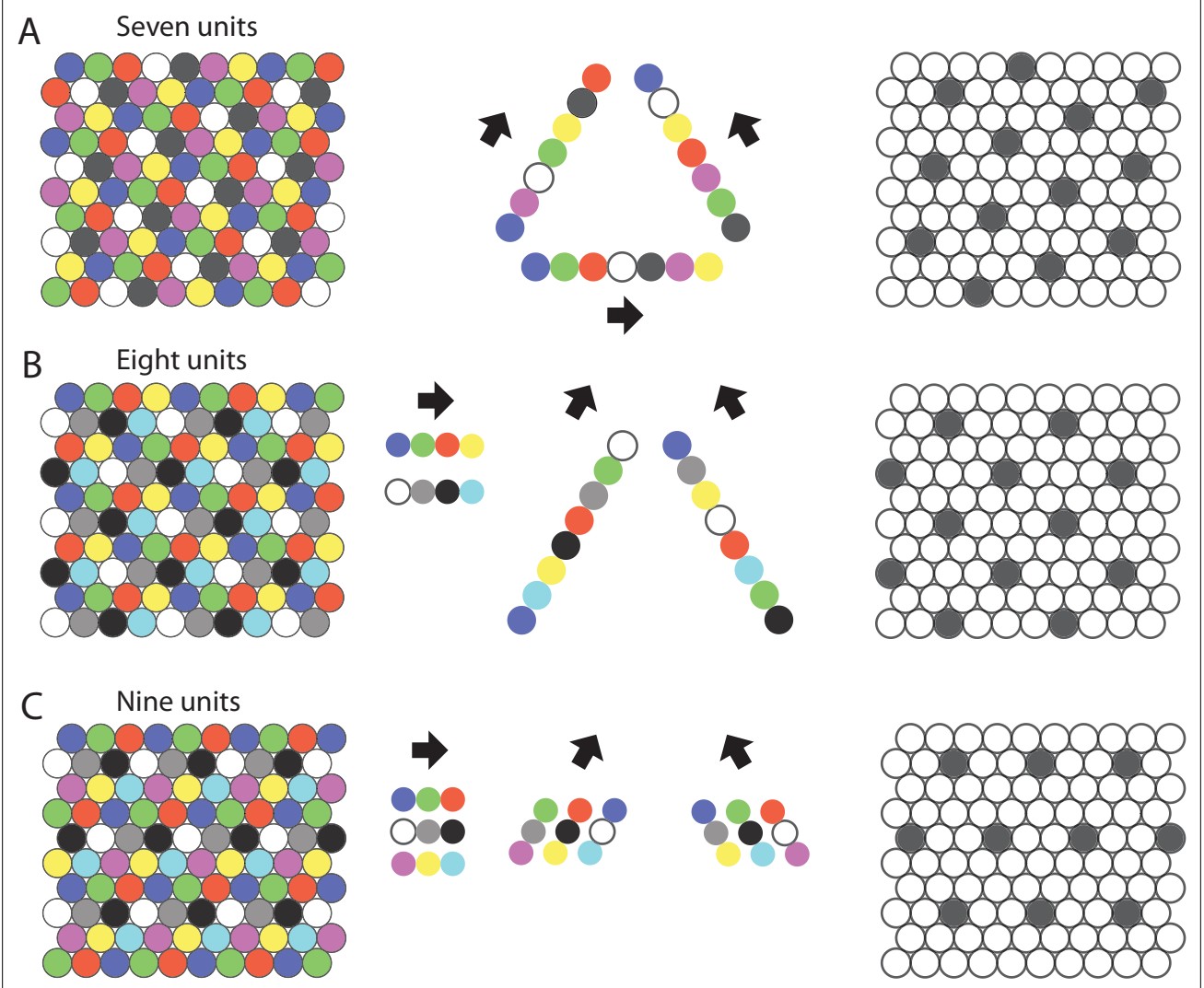

**Figure 2.** Repeating sequences of elements coding for trajectories result in lattice packing of firing fields. Left panels show firing fields of multiple distinct elements with non-overlapping firing fields arranged in a 2-D lattice packing (grid). Each grid provides a sequence code of trajectories. Mid panels show the sequences that code for the directions along the three major axes of the grid. Right panels highlight the firing fields of an individual element visualizing the type of lattice formed by the firing fields of one element. (**A**) An example of the minimal number of seven elements and resulting sequences with repeat length 7 along the three grid axes. The resulting firing fields of individual elements (referred to as a 'grid map' in animal experiments) are arranged on a hexagonal lattice rotated by 10.9° against a vertical axis. (**B**) An example of eight cells and sequences with repeat lengths 4 along one axis, and repeat length 8 along the two other axes. The resulting firing fields are arranged on a centered rectangular lattice. (**C**) An example of nine cells and sequences with repeat length 3 along all three grid axes, resulting in hexagonal lattice packing of individual elements' firing fields.

First, we will demonstrate that every sequence of elements along one of the three major axes of the hexagonal lattice (rows and diagonals) must be either a translation of an existent sequence in a parallel axis or consist of a disjoint set of elements.

To prove this statement, suppose that row A consists of elements $1, \ldots, k$ repeating in this order. Then any row that contains any element from $1, \ldots, k$ must contain the full repeat $1, \ldots, k$ by axiom 1. So any row containing any element from $1, \ldots, k$ is a translation of row A, and any element that does not contain them is disjoint from row A.

### Remark 12.3

We count the distance of a translation of a row A by starting at an element $a$, passing to the element diagonally down and to the right of element $a$, and counting spaces moved right from here. Translations must be by $0, 1, 2, \ldots, k-1$ spaces, where row A contains $k$ elements.

We note that if a row or diagonal of the hexagonal lattice is a translation of a neighboring row or diagonal, the translation must be by at least 2 and at most $n-3$, where $n$ is the repeat length. If the translation is by 0, we have one element firing adjacent to itself, which contradicts axiom 1. If the translation is by 1, say element $i$ has elements $j, i$ below and immediately to the right (south-east) of it, then $j, i$ is now coding for two distinct trajectories, namely north-west and east. This contradicts Axiom 1. If the translation is by $n-2$, or $n-1$, these are left-right reflections of translations by 1 or 0, respectively, and have the same issues.

Further, it must be true that sequence repeats in parallel rows or diagonals must be of the same length.

To illustrate this, assume two consecutive rows A and B have sequence repeats of different lengths, say row A consists of $1, \ldots, k$ and row B consists of $k+1, \ldots, k+s$, where $k < s$. Then, the element sequence $1, k+1$ will appear in one repeat and the element sequence $1, k+t$ will appear in the next, where $t$ is not equal to 1. This contradicts Axiom 1 (*Figure 3*) 25A1.

A final useful fact is that the length of the sequence repeat in any row or diagonal must divide the number of elements. As a result, if the number $n$ of elements is prime, the length of the sequence repeat in any row or diagonal must be of length $n$.

To prove this last fact, we will work by supposing the repeat length is less than $n$ in some dimension and showing that the repeat length must then divide $n$. Without loss of generality, assume it is the horizontal (west-east) direction, so that one row consists of $1, 2, \ldots, i$ repeating, for some $i < n$. Then, every row must consist of elements $1, 2, \ldots, i$ or be disjoint from the elements in other rows. Since every element will appear in exactly 1 of the distinct rows, the total number of elements must divide evenly into sets of i, one for each distinct row. So, the length of the repeat in each row must divide $n$.

If $n$ is prime, every row has repeat length dividing $n$. Since no row can have a repeat of length 1 by the work above, every row has repeat length $n$.

Next, we prove that the minimum number of trajectory-coding elements in 2-D space is 7 and describe the possible arrangements of 7 elements' firing fields in 2-D space that establish a sequence code of trajectories.

Given the requirement that each firing field is surrounded by 6 firing fields of distinct elements, the minimum number of elements providing a sequence code of trajectories in 2-D space is 7.

By our work above, the repeat length must be 7 in all directions. We may assume the elements are labeled such that row A consists of elements 1, 2, ..., 7 (*Figure 3B*). Hence, every row is a translation of row A. Since the translation must be at least two and at most four, we have a total of three options for how much row B is translated from row A. In *Figure 3B*, we show that translation by three yields a contradiction, so translation by two and four are the only possible options. Once rows A and B are determined, every other row is uniquely determined by Axiom 1 25A1.

There are thus exactly two possible arrangements of seven elements so that they form a sequence code for trajectories in 2-D space, up to relabeling of cells. Intriguingly, both possible arrangements of seven elements result in a hexagonal arrangement of firing fields of all distinct elements, and these arrangements are rotated against the wall of a rectangular enclosure by 10.9° (*Figure 3B*).

While the two solutions to a sequence code of trajectories with seven elements both result in a hexagonal lattice packing of firing fields, solutions with eight or more elements can result in other types of lattice packing (*Figure 1—figure supplement 1*), such as the oblique lattice or centered rectangular lattice that have also been observed in single element recordings of freely behaving animals (*Stensola et al., 2015*). Code to compute and visualize 'grid maps' of trajectory-coding sequence arrangements with up to 14 elements has been made publicly available to help the scientific community test some of our predictions against future datasets (*Gleeson et al., 2017*). Because (1) each row must either be a translation of the previous row or entirely disjoint from it, (2) the length of the repeat in each row is the same and divides the total number of elements, and (3) if one row is translated from the one above, the translation must be by at least 2 and at most n-3 (where n is the length of the repeating sequence in the previous row), we can classify all possible arrangements of trajectory coding sequences of elements for any particular number of elements. In *Table 2*, we list all possible

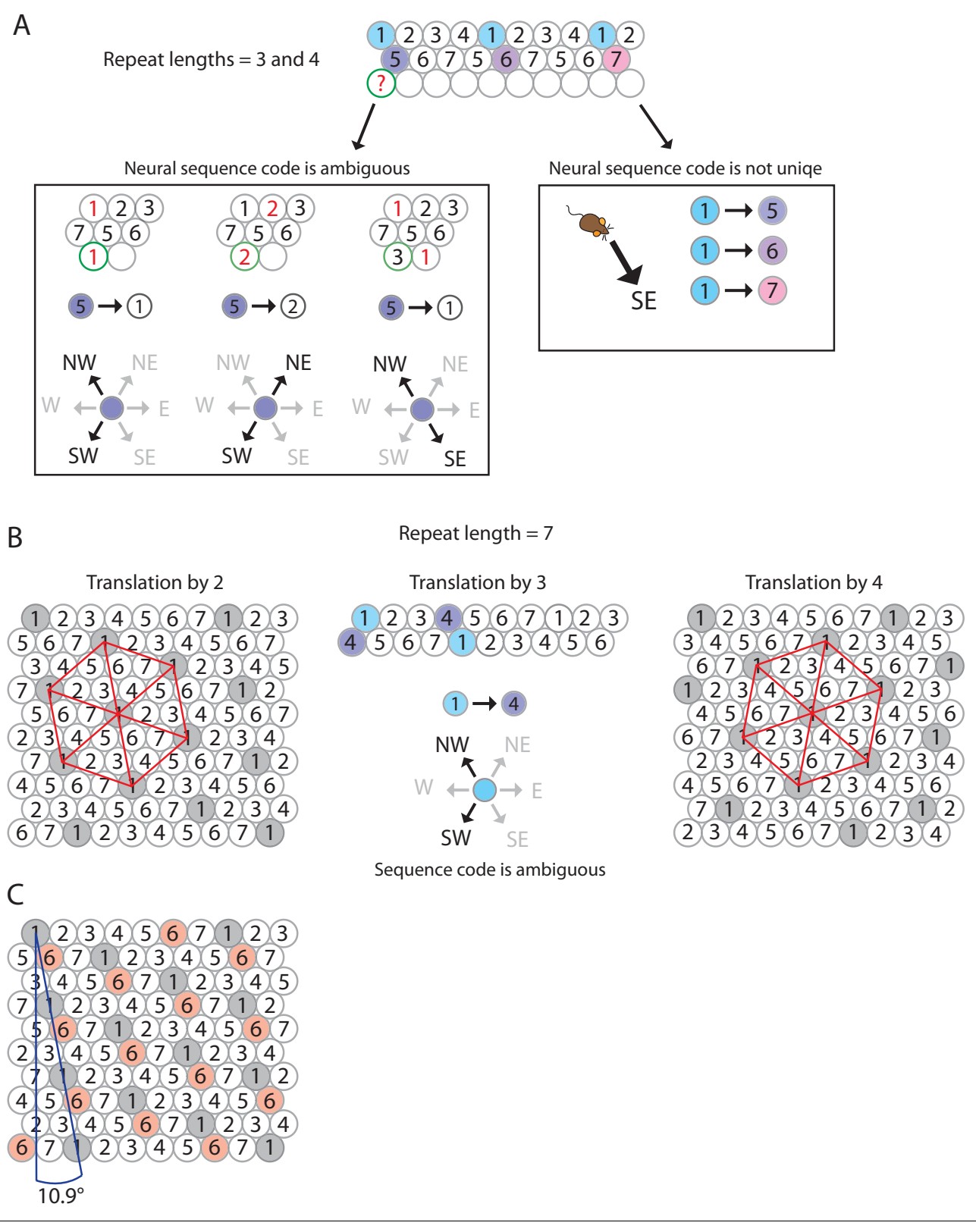

**Figure 3.** The minimum number of distinct elements providing a sequence code of trajectories in 2-D space is 7, and there are exactly 2 possible arrangements of elements up to relabeling. (**A**) Repeat lengths that are smaller than 7 result in a violation of a sequence code of trajectories because the code would be ambiguous and not unique. A red question mark indicates that no element can be found for this position that would not violate the sequence code. Red numbers indicate that activity of this element at the current position would violate sequence coding of trajectories due to

*Figure 3 continued on next page*

*Figure 3 continued*

ambiguity. Blue background color marks the sequential activity of two elements that violate the sequence code of trajectories (Definition 8) because the next active cell in the sequence is not uniquely determined by the currently active cell and the velocity vector associated with the current trajectory of an animal. (**B**) If the repeat length is 7, there are three potential translations of the sequence in row 1 to fill up row 2 (see Mathematical proof). Only two of those three translations result in an arrangement of elements (up to relabeling of elements) that creates a sequence code for trajectories in 2-D space. These two arrangements are mirror images of each other and imply that the firing fields of each individual grid element fall on the vertices of equilateral triangles, i.e., they form a hexagonal grid. (**C**) Both possible arrangements of firing fields imply that grid patterns of other grid elements have the same spacing and rotation, and only differ in spatial phase (compare grid patterns of element #1 and element #6). The smallest angle between one grid axis and the boundary of a rectangular enclosure is 10.9° if one row of sequences is aligned with one of the borders of the enclosure.

The online version of this article includes the following figure supplement(s) for figure 3:

**Figure supplement 1.** Quantification of grid field spacing and grid field size in grid cells obtained from mice.

**Figure supplement 2.** Grid field detection with filter threshold set to 31% of the peak firing rate.

ways to construct a sequence code composed of 7–12 trajectory-coding elements up to translation and rotation. Note that increasing the number of elements in a sequence code does *not* increase the spatial or angular resolution in the representation of a trajectory. Therefore, a code that uses more than the minimal number of seven elements has no functional benefit (see 'Discussion').

The grid pattern that we have found to emerge as the most parsimonious solution assuming axioms 1–4 (see 'Results'), that is, the solution to a sequence code of trajectories using a minimal number of elements, mirrors experimental results on grid elements from animal experiments as we demonstrate below.

## Sequence coding of trajectories predicts grid spacing and rotation of grid maps against a wall

The solution to sequence coding of trajectories predicts firing fields on the single-cell level that form a hexagonal lattice, as observed in grid cell data obtained from freely behaving rodents (*Hafting et al., 2005*; *Fyhn et al., 2008*) and crawling Egyptian fruit bats (*Yartsev et al., 2011*). The triangular structure of grid cells' firing maps (grid maps) has been characterized by three parameters, namely grid field size, grid spacing, and grid orientation (*Hafting et al., 2005*). Because the sequence code of trajectories model of grid cell firing implies a dense packing of firing fields, the spacing between two adjacent grid fields must change linearly with a change in field size. It follows that the ratio between grid spacing and field size is fixed. When using the distance between the centers of two adjacent grid fields to measure grid spacing and a diameter-like metric to measure grid field size, we can compute the ratio of grid spacing to grid field size as $\sqrt{7} \approx 2.65$ (see 'Methods'). We tested this prediction on a data set of n = 27 grid cells recorded in mice for a previously published study (*Dannenberg et al., 2020*). This data has been openly released in another study (*Sutton et al., 2024*). Quantifying field sizes requires identifying the borders of individual fields as a first step. Since the characteristic feature of a firing field is its firing rate, we chose to use a threshold for the firing rate to determine the field boundaries (see 'Methods'). Setting the field detection threshold to 31% of the peak firing rate optimized detection of grid fields in most grid cells (n = 25 out of 27 cells) (*Figure 3—figure supplement 1A*). We found that the ratio of the field spacing to a diameter-like metric of the field size was 2.39 ± 0.25 (mean ± SD; n = 25) (*Figure 3—figure supplement 1B*), which is only slightly smaller than 2.65, the upper bound predicted by the model. Note that the ratio of grid field spacing to field size observed in experimental data is expected to be smaller than the model-predicted ratio because of the many factors that result in out-of-field firing of neurons recorded in experiments, such as transient drifts of grid maps, path integration errors, conjunctive coding properties, and noise in experimental measurements (*Figure 3—figure supplements 1C and 2*).

Another testable prediction of our model is that the grid map associated with the most parsimonious solution using seven elements is rotated by 10.9° against the nearest wall of a rectangular enclosure (*Table 2*). This angle is well within the range of experimentally observed values. Concretely, *Stensola et al., 2015* reported a distribution of angles with a plateau between ~6 and ~12° (mean ± SD: 7.2 ± 3.5). Notably, the median of the angle to the nearest wall in a highly familiar as opposed to a novel environment was 9.8° (*Stensola et al., 2015*). Under the premise that a sequence of firing fields aligns with one of the geometric boundaries of the environment, the sequence code model explains that the grid pattern typically assumes one of only four distinct orientation configurations relative to

**Table 2.** List of possible grid maps and properties of these grid maps given the total number of elements participating in a trajectory-coding sequence in a rectangular environment.

For each individual element's grid map that has a non-zero angle to a border, a reflection or 90° rotation and reflection up to relabeling of the elements exist, and these additional possibilities are not included in this list.

| # cells | Repeat lengths* | Lattice type | Smallest angle to a border of a rectangular environment† |
|---|---|---|---|
| 7 | 7, 7, 7 | Hexagonal | 10.9° |
| | | | |
| 8 | 4, 8, 8 | Centered rectangular | 0° |
| | 4, 8, 8 | Oblique | 10.9° |
| | | | |
| 9 | 3, 3, 3 | Hexagonal | 0° |
| | 3, 9, 9 | Oblique | 10.9° |
| | 3, 9, 9 | Oblique | 13.9° |
| | 3, 9, 9 | Oblique | 16.1° |
| | | | |
| 10 | 5, 10, 10 | Oblique | 0° |
| | 5, 10, 10 | Oblique | 10.9° |
| | 5, 10, 10 | Oblique | 16.1° |
| | | | |
| 11 | 11, 11, 11 | Oblique | 6.6° |
| | 11, 11, 11 | Oblique | 10.9° |
| | 11, 11, 11 | Oblique | 16.1° |
| | | | |
| 12 | 6, 6, 6 | Hexagonal | 0° |
| | 3, 12, 12 | Rectangular | 0° |
| | 3, 12, 12 | Rectangular | 10.9° |
| | 3, 4, 12 | Oblique | 0° |
| | 3, 4, 12 | Oblique | 13.9° |
| | 4, 6, 12 | Oblique | 0° |
| | 4, 6, 12 | Oblique | 6.6° |
| | 4, 6, 12 | Oblique | 23.4° |

*Repeat lengths across the three major axes of the hexagonal lattice structure of densely packed firing fields.
†The smallest angle to a border of a rectangular environment assumes that one sequence of elements is arranged in parallel to one wall.

the environment (*Stensola et al., 2015*; *Stensola and Moser, 2016*). Concretely, the four orientation configurations arise when one row of grid fields aligns with one of the two sets of parallel walls in a rectangular environment, and each arrangement can result in two distinct orientations (*Figure 3B*). The sequence code model of grid cell firing thus provides a simple and intuitive answer to the otherwise puzzling observation that most grid field maps observed in freely behaving animals are rotated against the nearest wall of a rectangular enclosure.

## Sequence coding of trajectories predicts fragmentation of grid maps in 1-D space

Another experimental observation related to the spatial geometry of grid fields that can be explained by a sequence code of trajectories is the fragmentation of grid cell maps in a multicompartment

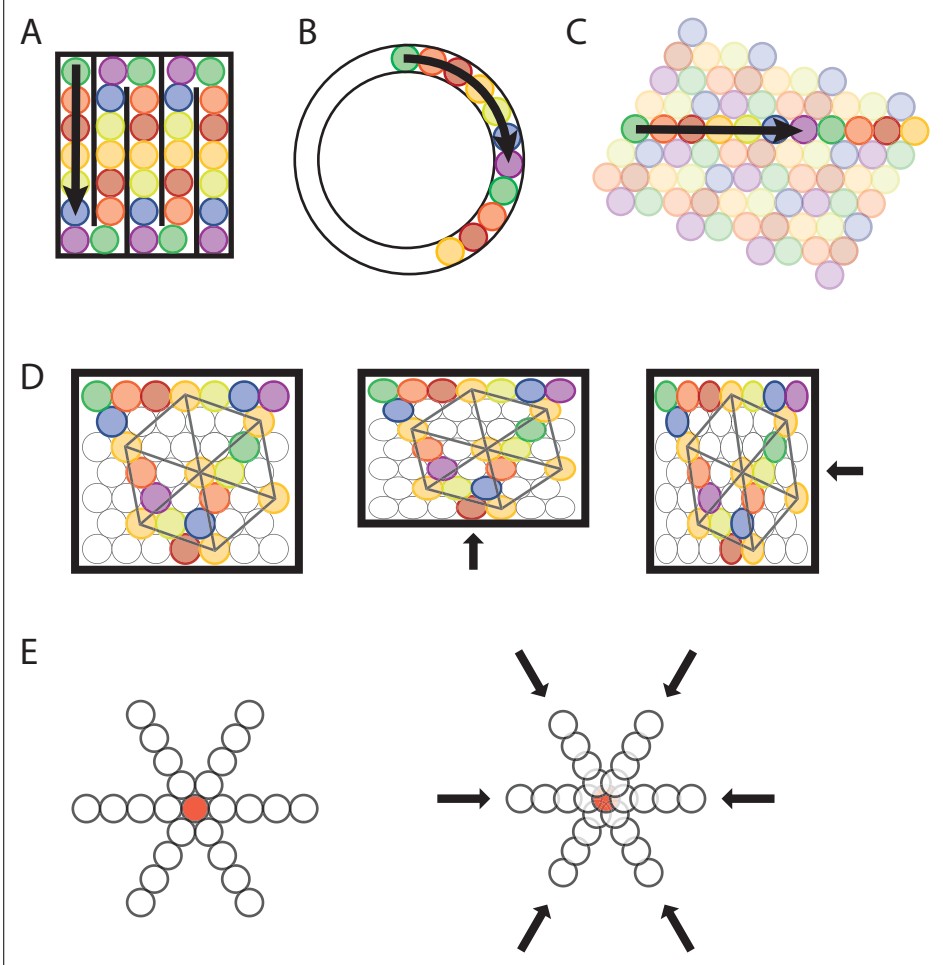

**Figure 4.** Transformation of a sequence code of trajectories from 2-D space to 1D space. (**A, B**) A sequence of seven different elements can code for a trajectory in a 1-D compartmentalized 'hairpin' maze (**A**) or a 1-D circular track (**B**). Note that the distance between firing fields would increase in the 1-D 'hairpin' maze and 1-D circular track compared to the distance between spatial firing fields in a 2-D environment. Also note that sequences could undergo a phase reset at behaviorally relevant points, for example, the turning points in the 'hairpin' maze (**A**). (**C**) The sequence of active elements in a 1-D environment can be interpreted as a cross-section of the trajectory sequence code in a 2-D space. (**D**) Anchoring of firing fields to environmental borders predicts parametric rescaling of an individual element's grid pattern when a familiar enclosure is deformed. Each color represents one grid element, each circle represents one firing field. For clarity, the complete set of firing fields of an individual element is shown only for one element (orange color). In addition, sequences are shown for all three major axes. Left panel, original maze configuration. Mid and right panel, the environment is compressed along the vertical or horizontal dimension resulting in a parametric deformation of firing fields and the grid pattern of individual elements' firing fields along the vertical or horizontal dimension. (**E**) Progressively faster advancement from the currently active element to the next active element in the sequence code of trajectories results in progressive decrease in grid spacing and thereby a local distortion of the grid map. Open circles represent firing fields of grid elements along the three major axes. The red-filled circle in the center represents a salient location such as a rewarded goal location toward which nearby grid fields gravitate.

environment (*Figure 4A–C*). The 2-D spatial periodicity in grid cell firing is replaced by one-dimensional (1-D) spatial periodicity if movement through space is restricted to 1-D trajectories along parallel alleys in a multicompartment maze (*Derdikman et al., 2009*). A sequence code of trajectories predicts such a fragmentation of grid cell maps because parallel trajectories would result in the same sequential activation of cells, and neural coding on a 1-D tract is most efficient when sequences are aligned with the movement direction (*Figure 4A*). Likewise, grid cells have been shown to path integrate distances on a 1-D circular track (*Jacob et al., 2019*), consistent with a sequence code of a trajectory

along a 1-D circular track (*Figure 4B*). Notably, *Jacob et al., 2019* report in their study that the field spacing of grid cells is increased in a 1-D circular track compared to a 2-D environment. The authors' explanation is that field spacing is increased due to the lack of visual cues. However, an alternative explanation based on the sequence model of grid cell firing is that the animal's trajectory on a linear track is represented by a sequence of cells (*Figure 4C*). Moreover, the sequence code of trajectories is consistent with an analysis of experimental data (*Pröll et al., 2018*) showing that firing fields of grid cells on a 1-D linear track are compatible with a slice through a 2-D hexagonal pattern. Specifically, a slice through a 2-D hexagonal firing pattern explains linear-track data if translational shifts of the pattern are allowed at turning points without a requirement of rotating or scaling the grid. In the context of the sequence code of trajectories, a translational shift of the grid pattern of a single cell is equivalent to re-anchoring the sequence code after the animal has turned around facing the opposite direction on the linear track. Such a translational shift or re-anchoring of the sequence code is consistent with experimental data showing differential spatial coding by place cells in the hippocampus for inbound and outbound running directions on a linear track (*Dombeck et al., 2010*).

Furthermore, grid cells have been shown to code for traveled distance, elapsed time, or a combination of distance and time when animals run in place on a treadmill (*Kraus et al., 2015*). The firing pattern of grid cells as time or distance cells is remarkably similar to the firing pattern of grid cells that emerges when animals navigate 1-D linear tracks after correcting for logarithmic expansion in grid field size over time. The fact that grid cells show repeating firing fields in other dimensions than physical space is consistent with a sequence code of trajectories in any type of dimension that has behavioral relevance for the animal. Experimental data by *Kraus et al., 2015* demonstrate parallel coding of different dimensions by different grid cells, suggesting individual grid cell sequences can provide independent codes of trajectories through different dimensions, thereby enabling parallel processing of different cognitive functions.

## Sequence coding of trajectories defined relative to landmarks or boundaries predicts rescaling and restructuring of grid maps with changes in landmarks and boundaries

A sequence code of trajectories does not need to be rigid but instead can be malleable and rescaled in response to changes in the environment as long as the start and end points of the sequences are anchored to salient environmental landmarks. Thus, sequence coding of trajectories by grid cells provides an explanation of distortions of the grid pattern that are frequently observed in animal studies. Experiments in freely behaving rodents demonstrate that stretching or compression of enclosed environments results in stretching or compression of the grid map in the rescaled dimension. For instance, stretching or compressing the borders of a familiar open-field recording arena resulted in an increase or decrease of the distance between firing fields in the rescaled dimension of the deformed enclosure (*Stensola et al., 2015*; *Barry et al., 2007*). Such distortions have been challenging to interpret under the assumption that grid cells provide a 'spatial metric' or 'coordinate system' for navigation. However, if we assume a sequence code with start and end points of trajectories anchored to landmarks or environmental borders, distortions of the grid along the rescaled dimension would be expected. The sequence code of trajectories would remain the same if trajectories are defined relative to the borders of the environment (*Figure 4D*). Moreover, experimental data in rats show that grid maps that have been established in two adjacent compartments separated by a wall within the same larger environment merge once the wall is removed (*Wernle et al., 2018*). Merging of the two grid maps appeared to happen instantly, resulting in local spatial periodicity and continuity between the two original maps (*Wernle et al., 2018*), consistent with the sequence code of trajectories by grid cells.

Scaling of grids has also been observed in the form of an expansion, that is, increased grid spacing between individual firing fields and increased field sizes, in response to novelty of the environment (*Barry et al., 2012*). Such scaling has also been reported in climbing rats foraging on a vertical wall (*Casali et al., 2019*). Both the scaling of grids in response to scaling the borders of an environment and the expansion of grid fields in response to novelty are inconsistent with the hypothesis that grid cells provide a spatial metric for navigation or path integration. However, an increase in grid spacing in response to a novel environment can be explained by a sequence code of trajectories reflecting an initially broader spatial resolution. Specifically, the spatial resolution in the trajectory code is

equivalent to the spacing between two adjacent spatial fields, and the spatial resolution is directly proportional to both the grid spacing and the field size. Initially, the overall layout of a novel space is likely more important than the fine-grained details. However, as the animal becomes more familiar with the environment, it is likely paying more attention to spatial details requiring a trajectory code with higher spatial resolution, resulting in a smaller grid map. While this hypothesis needs to be tested in future studies, experimental data show that changes in grid field size co-occur, though to a lesser extent, with changes in the size of place cell firing fields in response to novel environments (*Barry et al., 2012*; *Wilson and McNaughton, 1993*; *Karlsson and Frank, 2008*). In summary, sequence coding of trajectories is consistent with the experimentally observed scaling of grids in response to deformations of the borders of the environment and with compression or expansion of firing fields and grid spacing in response to novelty.

Furthermore, a sequence code of trajectories in 2-D space is consistent with experimental data showing that grid fields move toward goal locations (*Boccara et al., 2019*) or restructure their spatial firing maps to incorporate the location of a learned reward (*Butler et al., 2019*). These data support the hypothesis that grid cells do not provide a simple metric of space. Instead, the spatial firing pattern of grid cells is malleable in response to relevant contextual features. Such local distortions of global grid patterns by salient locations have recently been compared to spacetime distortions by blackholes (*Ginosar et al., 2023*). We propose that such warping of grid maps is caused by an increased probability that the currently active grid cell activates the next grid cell in the sequence of trajectory-coding cells, resulting in the center of firing fields moving closer together. A reduction of the spacing between spatial fields has been observed in hippocampal place fields, resulting in a backward shift with experience (*Mehta et al., 2000*; *Lee et al., 2004*; *Roth et al., 2012*; *Geiller et al., 2017*; *Dong et al., 2021*) due to NMDA receptor-dependent plasticity (*Ekstrom et al., 2001*). Under the premise that the grid map is anchored to the reward site, this would result in a progressive decrease in grid spacing toward the reward site, thereby causing local distortions in the grid map of a single grid cell (*Figure 4E*). A sequence code of trajectories by grid cells is thus consistent with malleable grid maps because the spacing between adjacent firing fields can change in response to local contextual features without compromising the sequence code. The same mechanism can explain shearing-induced asymmetry and multiple alignment solutions that have been shown in experimental data (*Stensola et al., 2015*).

## Multiple grids provide nearly continuous resolution in the coding of trajectories and predict grid cell properties on the population level

In addition to properties of grid cell firing on the single cell level, the definition of sequence coding of trajectories implies an intriguing property of grid cells on the population level, namely that each grid cell's collection of firing fields is geometrically congruent to any other grid cell's collection of firing fields. This property mirrors experimental data from single-element recordings in rodents showing that adjacent firing field grid patterns of different anatomically close grid cells differ only in phase but have the same orientation and scale (*Hafting et al., 2005*). Since the firing fields of distinct elements (i) do not overlap and (ii) discretize directions into six running directions only, we asked (i) how other running directions are represented and (ii) how partial transitions between firing fields can be accomplished. Consistent with experimental data (*Hafting et al., 2005*), the sequence code of trajectories model of grid cell firing accounts for smooth transitions between firing fields and the representations of more than six running directions by using multiple grids of seven elements each, where each element has the same field size. However, the firing fields are shifted in phase. It follows from Axiom 3 that one needs to add three grids composed of the same number of elements as the original grid and shifted in phase such that their field centers fall exactly between the firing fields along one major axis of the original grid (*Figure 5A and B*). Consequently, the firing fields of all distinct elements of the compound grid must again fall on a hexagonal lattice.

Two key conclusions from this theoretical result are that (i) an arbitrarily smooth transition between grid fields/grid cells becomes possible during navigation with an arbitrarily large number of grid cells (organized in fundamental grids of seven elements with non-overlapping firing fields), and (ii) traveling directions can be represented with an arbitrarily high angular resolution (see also *Video 1* of the sequence model associated with this study). This is consistent with experimental data on simultaneously recorded multiple grid cells with overlapping firing fields and consistent with data showing that grid cell dynamics can be represented with a toroidal manifold (*Figure 5C*).

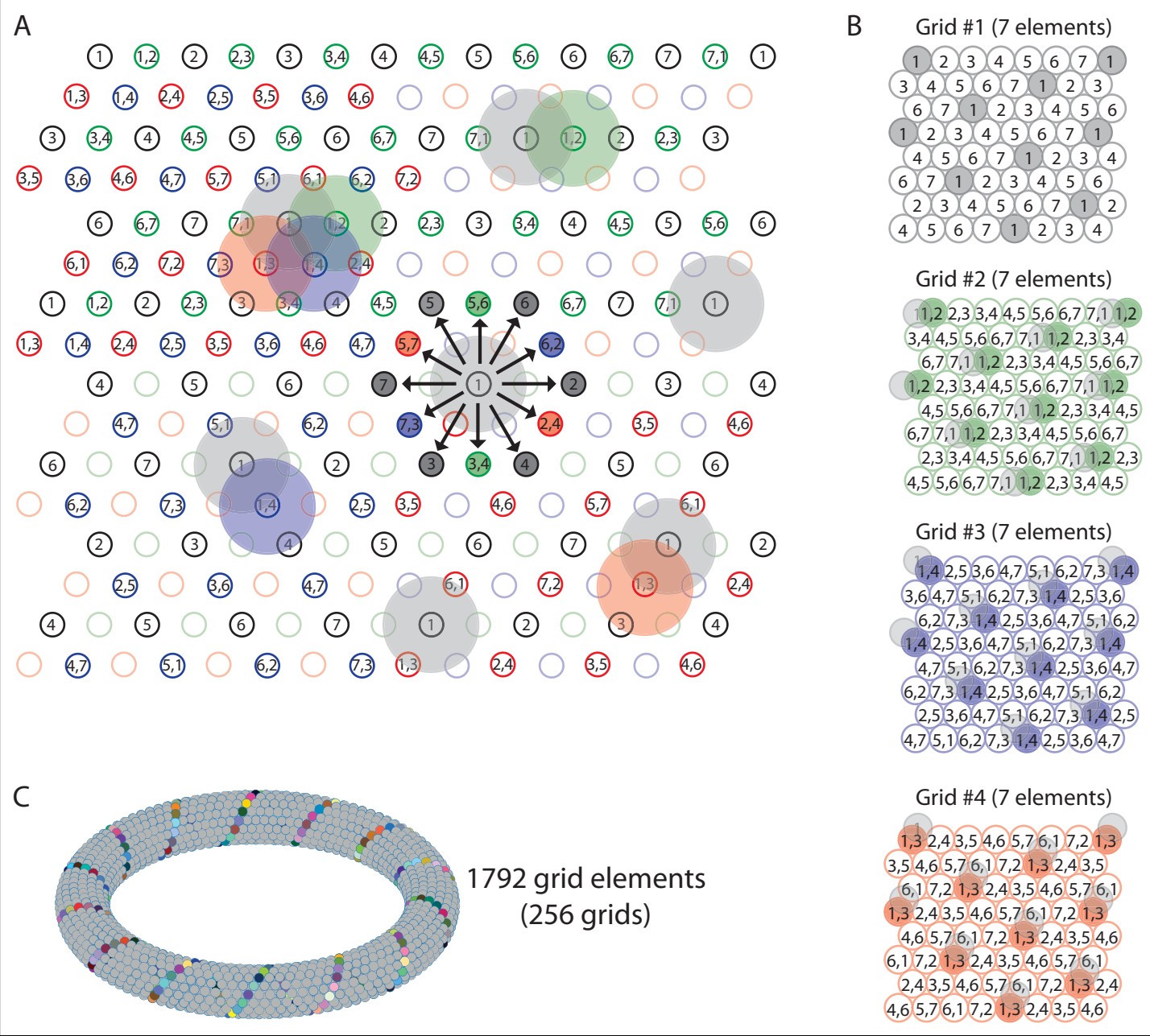

**Figure 5.** Multiple grids provide nearly continuous resolution in the coding of trajectories. (**A**) The circles with numbers show the *centers* of the grid fields from four grids, each composed of seven distinct elements (total of 28 grid elements). Shaded areas represent the firing fields of individual elements. Elements with non-overlapping firing fields form 'grids', shown in black, green, blue, and red colors. Each number within a circle identifies a distinct element. If elements represent cell assemblies, a combination of numbers (e.g., '1,2') identifies an element whose cellular composition is an overlap between two elements with adjacent firing fields. For example, element '1,2' is composed of cells that are part of elements '1' and '2', resulting in an overlap of firing fields. For visual clarity, not all element numbers are shown. (**B**) Grids #2, #3, and #4 are phase-shifted along one of the three major axes of grid #1. The emerging grid map of an individual element's firing fields is highlighted in color for one grid element in each grid. The phase-shift of each grid is indicated by showing the transparent grid pattern of grid #1. (**C**) When more grid elements are added, the elements can be represented in 3D neural space as a neural manifold, shown here for a total of 1792 grid elements forming 256 grids of non-overlapping firing fields. The colored cells represent one diagonal axis across rows of non-repeating elements if plotted in the 2-D plane as shown in (**A**).

The online version of this article includes the following figure supplement(s) for figure 5:

**Figure supplement 1.** Grid cell modules provide a pyramidal parametric sampling of 2-D space enabling simultaneous multiscale representation of space.

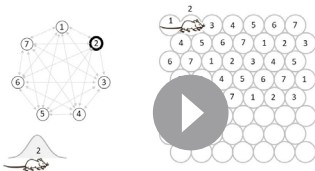

**Video 1.** Local path integration in a sequence code of trajectories begets hexagonal firing of grid cells. https://elifesciences.org/articles/96627/figures#video1

Moreover, experimental data have shown that grid cells cluster into autonomous modules with distinct scale, orientation, symmetry, and theta frequency modulation (*Stensola et al., 2012*). If we assume that grid cells emerge from a sequence code of trajectories, adjacent firing fields in that sequence would provide a discrete sequence of sampling points in the coding of trajectories, and the distance between the centers of two adjacent firing fields would determine the spatial resolution. Doubling the number of sampling points, that is, doubling the number of firing fields per unit area would double the spatial resolution (*Williams, 1983*). Conversely, dividing the number of firing fields per unit area in half would halve the spatial resolution. To double or halve the spatial resolution in 2-D space, the number of firing fields per area needs to double or be cut in half, respectively (*Figure 5—figure supplement 1*). Because firing fields are densely packed, doubling the number of firing fields per unit area in 2-D space would result in dividing the distance between two adjacent firing fields by a factor of √2. This is exactly what has been observed in animal experiments (*Stensola et al., 2015*).

## Discussion

### A teleological cause for the existence of grid cells

In this study, we prove mathematically that spatial periodicity in grid cell firing emerges as the only possible solution to provide a code of trajectories in 2-D space by cell sequences. Within the space of all possible solutions, hexagonal symmetry in grid cell firing emerges as the most parsimonious solution with respect to the required number of neurons. The hexagonal firing pattern of grid cells in 2-D space is arguably their most intriguing property. Yet, despite decades of experimental and theoretical research, the fundamental nature of the computational problem solved by grid cells has remained largely elusive. Previous studies on grid cells have been fruitful in describing potential functions of the grid cell system such as providing a population code for spatial location (*Sreenivasan and Fiete, 2011*; *Mathis et al., 2012*) and in describing functional properties of grid cells and grid cell firing at the level of neuroanatomy, connectivity, neurophysiology, and circuit dynamics. Yet, a specific and intuitive answer to why grid cells have evolved has remained elusive. What is the computational advantage of grid cells over other potential mechanisms or algorithmic implementations in the context of a computational goal? Previous investigations of grid cell function (*Zilli, 2012*; *Couey et al., 2013*; *Zutshi et al., 2018*) based on mechanistic models such as the oscillatory interference (*O'Keefe and Burgess, 2005*; *Burgess et al., 2007*; *Giocomo et al., 2007*; *Hasselmo et al., 2007*; *Burgess, 2008*) or continuous attractor models (*Fuhs and Touretzky, 2006*; *Burak and Fiete, 2009*; *Shipston-Sharman et al., 2016*) do not address the computational goal of or teleological cause for the existence of grid cells. Other studies have used normative models to demonstrate that grid patterns can emerge to optimize the coding of space by grid cells (*Wei et al., 2015*; *Mathis et al., 2015*) or to optimize path integration (*Sorscher et al., 2023*; *Banino et al., 2018*; *Cueva and Wei, 2018*). However, these normative models fall short in demonstrating that grid cells are necessary for the assumed normative functions and often rely on hidden architectural, hyperparameter, and constraint choices to obtain grid cell firing patterns (*Schaeffer et al., 2022*) as discussed under the next section 'Normative models of grid cell firing'.

We therefore deviated from these traditional approaches of investigating the nature of grid cell properties and asked the question: Is there a brain function that could either not be performed at all or would be substantially more costly or difficult to perform in the absence of grid cells? To constrain our search for such a function, we assumed that any coding mechanisms that operates on spatial firing requires cell sequences as a fundamental neural syntax in the brain (*Buzsáki, 2010*; *Buzsáki et al., 2022*; *Zutshi et al., 2017*). Recent studies provided compelling evidence for sequential activity of neurons representing spatial trajectories. In particular, *Gardner et al., 2022* demonstrated that the

sequential activity of hundreds of simultaneously recorded grid cells in freely foraging rats represented spatial trajectories. Complementary preliminary results indicate that grid cells exhibit left-right-alternating 'theta sweeps', characterized by temporally compressed sequences of spiking activity that encode outwardly oriented trajectories from the current location (*Vollan et al., 2024*).

The concept of sequential grid cell activity extends beyond spatial coding. In various experimental contexts, grid cells have been shown to encode non-spatial variables. For instance, in a stationary auditory task, grid cells fired at specific sounds along a continuous frequency axis (*Aronov et al., 2017*). Further studies revealed that grid cell sequences also represent elapsed time and distance traversed, such as during a delay period in a spatial alternation task (*Kraus et al., 2015*). Similar findings were reported for elapsed time encoded by grid cell sequences in mice performing a virtual 'Door Stop' task (*Heys and Dombeck, 2018*).

Additionally, spatial trajectories represented by temporally compressed grid cell sequences have been observed during sleep as replay events (*Ólafsdóttir et al., 2016*; *O'Neill et al., 2017*). Collectively, these studies demonstrate that sequential activity of neurons within the MEC, particularly grid cells, consistently encodes ordered experiences, suggesting a fundamental role for temporal structure in neuronal representations.

The theoretical underpinnings of grid cell activity coding for ordered experiences have been explored previously by *Rueckemann et al., 2021*, who argued that the temporal order in grid cell activation allows for the construction of topologically meaningful representations, or neural codes, grounded in the sequential experience of events or spatial locations. However, while Rueckemann et al. argue that the MEC supports temporally ordered representations through grid cell activity, our findings suggest an inverse relationship: namely, that grid cell activity emerges from temporally ordered spatial experiences. Additional studies demonstrate that hippocampal place cells may derive their spatial coding properties from higher-order sequence learning that integrates sensory and motor inputs (*Raju et al., 2024*) and that hexagonal grids, if assumed a priori, optimally encode transitions in spatiotemporal sequences (*Waniek, 2018*).

Together, experimental and theoretical evidence demonstrates the significance of sequential neuronal activity within the hippocampus and entorhinal cortex as a core mechanism for representing both spatial and temporal information and experiences.

Assuming that neural coding of trajectories in 2-D space is important for animal navigation and implemented by neural sequences, we show that grid cells are the most effective solution to coding of trajectories by neural sequences. We demonstrate that seven cells are sufficient to provide an unambiguous code for trajectories in 2-D space and that the only two possible solutions with seven cells result in firing fields that fall on a hexagonal grid. Notably, these two solutions result in grid maps that are mirror images of each other up to relabeling the cells. Any other solution to the problem would require at least eight cells, an increase in the number of cells forming a grid by more than 14%. Note that an increase in the number of cells within one grid would not result in an increase in spatial or angular resolution in the coding of trajectories. It can therefore be reasoned that performing the same computational function with the minimum number of cells forming a grid is evolutionary advantageous because it conserves space, cellular material, energy, and resources (*Laughlin and Sejnowski, 2003*). A solution to a sequence code of trajectories that builds on fundamental units of only seven grid cells compared to a larger number of cells would then be expected to be most frequently adopted. We therefore argue that we have provided a likely teleological cause for the existence of grid cells and an answer to the fundamental question of why grid cells have emerged in brains of navigating animals, namely that grid cells are the most parsimonious solution to sequence coding of trajectories in 2-D space.

## Comparison to normative models of grid cell firing

Normative models generally fall into two classes: one that uses artificial neural network models (*Sorscher et al., 2023*; *Cueva and Wei, 2018*; *Banino et al., 2018*) and one that uses a mathematical or analytical approach (*Waniek, 2018*). Grid-like firing has been found to emerge in artificial neural networks when the models were trained to perform path integration under simple biologically plausible constraints. The emerged grid cells then endowed agents with the ability to perform vector-based navigation (*Sorscher et al., 2023*; *Banino et al., 2018*). While these models demonstrate that path integration is an important driving force in the generation of grid cells, the mechanism

that is generating grid cell firing patterns remains obscure due to the untransparent nature of neural networks. Moreover, the results often depend on specific hyperparameter choices to explain the emergence of grid-like elements under anatomical constraints (see *Schaeffer et al., 2022* for a discussion). In contrast, we have provided mathematical proof that spatial periodicity of firing fields emerges from a sequence code of trajectories in 2-D space and that the most parsimonious solution with only seven cells results in hexagonal lattice packing.

## Sequence coding of trajectories by grid cells provides a mechanistic explanation for how grid cells serve path integration and can support memory-guided navigation

If the step-by-step advancements from the currently active cell to the next cell in the grid cell sequence are caused by integrating velocity signals, grid cells perform path integration. The ensuing grid cell sequence is then the result of the path integration and can be interpreted as a sequence code by a downstream reader, potentially the hippocampus. While the sequence code of trajectories-model of grid cell firing is agnostic about the neural mechanisms that implements the sequence code, one plausible implementation is a continuous attractor network (*McNaughton et al., 2006*; *Burak and Fiete, 2009*). Interestingly, a sequence code of trajectories begets conformal isometry in the attractor network, that is, a trajectory in neural space is proportional to a trajectory of an animal in physical space. Different grid cell modules integrate velocity signals at different (spatial) scales that are determined by the distance between the grid field centers of adjacent cells in the sequence. A sequence code of trajectories by grid cells therefore provides a mechanistic explanation for the functional role of grid cells in path integration. The model described in this article can be implemented with temporal coding (spiking neurons) or rate coding, or even with binary activity states of neurons, where neurons are activated sequentially as a function of the animal's velocity. Previous modeling work has shown that circuit mechanisms using grid cells in combination with speed-modulated head direction cells and hippocampal place cells can provide a substrate for episodic encoding and retrieval of spatio-temporal trajectories (*Hasselmo, 2009*). Furthermore, grid cell sequences have been shown to be useful in simulations of new goal-directed trajectories, where grid cells were used in a network of head direction cells, hippocampal place cells and persistent spiking cells to plan forward trajectories through the environment that search for place cells near a goal location (*Erdem and Hasselmo, 2014*). Recent experimental data demonstrate that temporally coordinated entorhinal inputs drive hippocampal sequences to perform memory-guided navigation (*Liu et al., 2023*) and that temporally structured population activity of grid and place cells represents local trajectories essential for goal-directed navigation and planning (*Chaudhuri-Vayalambrone et al., 2023*).

As part of the proof that a trajectory code by cell sequences begets spatially periodic firing fields, we proved that the centers of the firing fields must be arranged in a hexagonal lattice. This arrangement implies that the neural space is a conformally isometric embedding of physical space, so that local displacements in neural space are proportional to local displacements of an animal or agent in physical space, as illustrated in *Figure 5*. This property has recently been introduced in the grid cell literature as the conformal isometry hypothesis (*Xu et al., 2024*; *Schøyen et al., 2024*). Strikingly, (*Schøyen et al., 2024*) arrive at similar if not identical conclusions regarding the geometric principles in the neural representations of space by grid cells.

It would provide an advantage to animals if they could plan trajectories toward goals on different spatial scales to 'zoom in' or 'zoom out' on a cognitive map of their environment. We therefore proposed that each grid module represents trajectories at progressively lower or higher resolution to enable planning of trajectories across different spatial scales. This approach is analogous to mip mapping known from computer graphics, where an image is sampled by a stack of images, each of which is represented with half the resolution of the previous to increase rendering performance (*Williams, 1983*).

A similar idea has previously been implemented in a scale-space model of grid cells (*Waniek, 2020*). It remains to be tested experimentally whether grid cell modules primarily serve the coding of spatial location or a multiscale representation of spatial trajectories. Furthermore, it remains to be shown why the scaling factor used in the grid cell system is 2 and not smaller or larger than 2. Interestingly, a scale ratio of 2 is commonly used in computer vision, specifically in the context of mipmapping and Gaussian pyramids using Gaussian filter kernels to render images across different spatial scales

while allowing a smooth and balanced transition between successive levels of an image pyramid (*Burt and Adelson, 1983*; *Lindeberg, 2008*). Examining whether larger factors lead to an excessive loss of detail between spatial scales, while smaller factors fail to sufficiently reduce the spatial scale to justify additional computational levels, represents a promising direction for future research.

The grid cell model presented in this study assumes complete tiling of space and equal representation of traveling direction. If this assumption was relaxed, there would be gaps in the representation of space and some traveling directions would be represented with larger or smaller fields, hence with smaller or larger angular resolution, respectively (see *Figure 1F and G*). However, path integration performed by animals is far from being perfect, and it would be of interest for future work to investigate whether non-ideal tiling of space by grid cells may be one of many factors contributing to errors in path integration (*Allen et al., 2014*; *Gil et al., 2018*). One potential mechanism to compensate for non-ideal tiling of space is population coding. Grid cells are part of grid cell assemblies, and the population activity of grid cell assemblies can be interpreted by a reader mechanism as a robust population code.

Notably, the trajectory code itself does *not* require anchoring to a reference frame to perform local path integration. Because of the local nature of the trajectory code, path integration can be performed locally without the emergence of a global grid pattern. This has been shown experimentally in mice performing a path integration task where changes in the location of a task-relevant object resulted in translations of grid patterns in single trials (*Peng et al., 2023*). Although no global grid pattern was observed because the reference frame was not fixed in space, grid cells performed path integration locally within the reference frame defined by the moving task-relevant object, and grid patterns were visible when the changes in the references frames were accounted for in computing the rate maps.

Sequence coding of trajectories is not limited to physical space but could be useful for cognitive functions such as working memory or episodic memory that connects events into a cohesive story, consistent with experimental data demonstrating a grid-like code of conceptual knowledge space in humans (*Constantinescu et al., 2016*). A sequence code of trajectories by grid cells can thus explain data from human subjects that link reduced grid-like activity with path integration deficits observed in Alzheimer's disease risk carriers (*Kunz et al., 2015*; *Bierbrauer et al., 2020*), and data from rodent experiments demonstrating that grid cell firing is reduced in a rodent animal model of Alzheimer's disease (*Ying et al., 2022*; *Ying et al., 2023*).

## Sequence coding of trajectories in 3-D space

There is no reason to assume that the sequence code of trajectories described in this study is restricted to 2-D space. We have discussed in previous paragraphs how a sequence code of trajectories by grid cells can code for trajectories in 1-D space. In principle, such a sequence code can be expanded to three-dimensional (3-D) or higher dimensional spaces. However, there are multiple equally optimal possibilities to generate a close-packed arrangement of firing fields in 3-D space (*Stella and Treves, 2015*; *Weisstein, 2025*), where each sphere touches 12 neighboring spheres. Two examples of lattice packings in 3-D that achieve this kissing number are the face-centered cubic and hexagonal closely packed arrangements, which both have layers arranged in hexagonal lattices but stack the layers differently (*Conway and Sloane, 1999*). A model based on a sequence code of trajectories by grid cells would therefore predict larger variability in spatial periodicity of grid cell firing in 3-D space compared to 2-D space. It remains to be determined whether specific solutions to cell sequence coding of trajectories in 3-D space could result in local or global symmetries. The distribution of grid cell firing fields in rats exploring a 3-D volumetric space is irregular (*Grieves et al., 2021*). However, since sequences need to repeat, cell sequence coding of trajectories in 3-D space predicts some degree of local order in the grid map of a single grid cell. Consistent with this prediction, experimental data show that grid cells recorded in flying bats exhibit fixed local distances between firing fields despite the lack of a global lattice arrangement of firing fields (*Ginosar et al., 2021*). Taken together, these experimental data are consistent with a sequence code of trajectories that expands to 3-D space, even though this sequence coding may not need to result in a globally symmetric arrangement of the firing fields of a single cell. Based on data from animals navigating in 3-D environments, it has recently been proposed that the characteristic hexagonal firing pattern of grid cells is a "by-product of whatever process causes the cells to fire in spatially discrete regions of uniform size" (*Jeffery, 2023*). This study demonstrates that the said process is a neural sequence code of trajectories.

## Limitations of the study

This study demonstrates a teleological cause for the existence of grid cells, namely sequence coding of trajectories in 2-D. The goal of this study was thus not to provide a mechanistic model of grid cells or biologically detailed model of the connections between neurons that can mechanistically account for grid cell firing. The model thus remains agnostic about the specific mechanistic implementation as to how a velocity signal is integrated in entorhinal circuits to move activity from one neuron in the sequence to the next neuron in the sequence as a function of the animal's velocity and makes no predictions about connectivity between neuron types and the connection strengths.

While we have demonstrated that the ratio of grid field spacing to grid field size in an experimental data set of grid cells obtained from mice is consistent with the prediction made by the trajectory code by neural sequences model of grid cell firing, we acknowledge that future studies should test this model prediction on larger data sets obtained from other species and recorded across different grid cell modules.

# Methods

**Key resources table**

| Reagent type (species) or resource | Designation | Source or reference | Identifiers | Additional information |
|---|---|---|---|---|
| Software, algorithm | MATLAB | The MathWorks | RRID:SCR_001622; version: 2021a | |
| Software, algorithm | | Repository: GitHub; reference: this paper | https://github.com/dannenberglab/grid-cell-sequences/blob/main/get_gridMaps.m copy archived at *Dannenberg, 2025* | MATLAB code to plot grid maps that emerge because of sequence coding of trajectories |
| Software, algorithm | | Repository: GitHub; reference: this paper | https://github.com/dannenberglab/grid-cell-sequences/blob/main/plot_torus.m copy archived at *Dannenberg, 2025* | MATLAB code to plot toroidal manifold of neural grid cell space from multiple grid units (related to *Figure 5C*) |
| Software, algorithm | | Repository: GitHub; reference: this paper | https://github.com/Hippocampome-Org/gridcell_metrics | MATLAB code to quantify grid field spacing and grid field size from experimental data on grid cells |

## Math notations

A *theorem* is a mathematical statement that can be proved using logical deduction from previously known results.

A *proof* is a logical argument using known results to generate new mathematical statements. The purpose of a proof is to convince the reader that the result follows from known results.

A *plane* is a flat surface extending infinitely far in two dimensions. Also referred to as $R^2$, since it can be assigned coordinates of the form (*a, b*) where *a* and *b* are real numbers.

A *Euclidean metric* is the usual notion of distance in the plane. Two points with coordinates $(x_1, y_1)$ and $(x_2, y_2)$ have distance $\sqrt{(x_1 - x_2)^2 + (y_1 - y_2)^2}$.

*Adding points* means adding their coordinates, *subtracting one point from another* involves subtracting their coordinates.

A *bijective map* is a correspondence between two sets such that every point in one set is matched with exactly one point in the other set.

A *geometric figure* is a geometric object with a shape in the plane.

We work in $R^2$ (the plane), with the Euclidean metric.

## Quantification and statistical analysis

### Prediction of ratio between grid field spacing and grid field size

The grid field spacing $s$ was computed as a multiple of the diameter $d$ of a circular grid field for a grid unit with seven cells. Using the Pythagorean theorem, we can write

$$s^2 = 2^2 + (2 * cos(30))^2$$

$$s^2 = 4 + (\sqrt{3})^2$$

$$s^2 = 7$$
$$s = \sqrt{7}$$

## Quantification of grid field spacing and grid field size in experimental data

Since the characteristic feature of a firing field is its firing rate, we chose to use a threshold for the firing rate to determine the field boundaries. Notably, choosing the threshold too low will result in firing fields that are too large and that will merge with each other, thus not allowing identification of individual fields. However, choosing the threshold too high will result in measurements of field sizes that are too small. To avoid bias regarding the predicted value, the person performing the analysis was blinded towards the value predicted by the sequence model of grid cells.

## Field detection software design

Software was created to automatically detect grid cell fields. Those detected fields are used to find the mean field size and spacing for each cell. Open source code for the software is available at https://github.com/Hippocampome-Org/gridcell_metrics (*Sutton et al., 2025*; copy archived at *Sutton, 2025*). The code includes data for all cells used for analyses in this article and allows a user to reproduce the field detection results. The software detects grid fields by the use of a threshold to filter out out-of-field firing. Out-of-field firing is approximated by firing at a location with a firing rate below a specified percentage relative to the peak firing rate given an animal location. The code performs grid field detection using the spatial autocorrelogram of the cell's firing rate map. The firing rate map was generated by dividing the open-field environment into 3 cm spatial bins and computing, for each spatial bin, the occupancy-normalized spiking rate as described in *Dannenberg et al., 2020*. Rate maps were then smoothed by a 3 cm wide 2-D Gaussian kernel.

The objective of the software is to only analyze the most central field and the six closest fields surrounding it to avoid complications with analyzing fields that contact borders of a plot and have part of their areas missing due to their fields extending past the border. Other fields are detected but not saved for analyses. The DBSCAN clustering algorithm from MATLAB's (mathworks.com) Statistics and Machine Learning Toolbox is used to automatically separate the field data into clusters. Each cluster represents a detected field area. The field centers are computed as the centroids of the detected clusters. The positions of centroids are computed by calculating the mean of all pixels on the x-axis and y-axis of a field cluster and taking the mean x-axis and y-axis values as the coordinates of the centroid for each field. The area of each field is calculated as the number of pixels in the field. The spacing between fields is calculated as the Euclidean distance between field centroids. The mean spacing value is computed from six measurements of the spacings from the center field to each of the six surrounding fields. The grid field size is computed as the mean of a diameter-like metric from each field's area. The diameter-like metric $d$ uses the formula for finding a circle diameter given its area, namely $d = 2\sqrt{(area/\pi)}$. The ratio of field spacing to field size is then computed as the field spacing divided by the field size.

## Assessment of field detection software performance

The grid field detection algorithm performed best (measured as the percentage of cells where grid fields could be detected) with the threshold set to 31% (*Figure 2*, *Figure 3—figure supplement 1B*). The evaluation of performance was based on data from 27 grid cell recording files that were a part of a previously published study by *Dannenberg et al., 2020*. Only recording files from baseline conditions (the 'Light' condition in *Dannenberg et al., 2020*) were analyzed. Grid cells that met a minimal grid score threshold of 0.19 were included. The grid score was computed as described in *Dannenberg et al., 2020*. Effective field detection by the software was confirmed by the experimenter as (1) the intended number of fields being detected (center field in plot and six closest to it) and (2) the fields being detected without merging of grid fields compared to visual inspection of the cells' firing rate map. Grid cells that did not have effective field detection by the software were not included in ratio measurements. The software-computed size of grid fields is a function of a threshold parameter that was constant for all cells and was chosen to optimize performance of the software such that the number of grid cells for which the software-detected grid fields was maximal. Since the threshold was constant for all cells, field sizes could appear larger or smaller relative to estimated field sizes based on

visual inspection of rate maps. However, this was not determined to be a reason to exclude a cell for analysis with the threshold parameter that detected those sizes. Evaluating what size is large or small is subjective, and such a subjective evaluation was avoided to benefit the ability to reproduce results. Small field detection at some thresholds is a known issue that can be addressed further in future work.

## Acknowledgements

This work was supported by the National Institute of Neurological Disorders and Stroke of the National Institutes of Health, grant numbers R00NS116129 to HD and R01NS39600 to GAA; and by the National Science Foundation, grant number 2424326 to RRG, GAA, and HD. We thank Harrison Bray for helpful conversations about lattice packings, and Xihui Zheng for help with graphical illustrations. We thank Michael E Hasselmo for sharing data with us and for comments on the manuscript. We thank Marc W Howard for helpful comments on the manuscript.

## Additional information

### Funding

| Funder | Grant reference number | Author |
|---|---|---|
| National Institutes of Health | R00NS116129 | Holger Dannenberg |
| National Institutes of Health | R01NS39600 | Giorgio A Ascoli |
| National Science Foundation | 2424326 | Rebecca RG<br>Giorgio A Ascoli<br>Holger Dannenberg |

The funders had no role in study design, data collection and interpretation, or the decision to submit the work for publication.

### Author contributions

Rebecca RG, Formal analysis, Funding acquisition, Investigation, Writing – original draft; Giorgio A Ascoli, Supervision, Funding acquisition, Investigation, Writing - review and editing; Nate M Sutton, Software, Formal analysis, Methodology, Writing – original draft; Holger Dannenberg, Conceptualization, Supervision, Funding acquisition, Investigation, Writing – original draft

### Author ORCIDs

Rebecca RG http://orcid.org/0000-0002-7700-4312
Giorgio A Ascoli http://orcid.org/0000-0002-0964-676X
Nate M Sutton https://orcid.org/0000-0002-4424-3886
Holger Dannenberg https://orcid.org/0000-0002-0340-0128

Reviewer #1 (Public review): https://doi.org/10.7554/eLife.96627.3.sa1
Reviewer #2 (Public review): https://doi.org/10.7554/eLife.96627.3.sa2
Reviewer #3 (Public review): https://doi.org/10.7554/eLife.96627.3.sa3
Author response https://doi.org/10.7554/eLife.96627.3.sa4

## Additional files

### Supplementary files
MDAR checklist

### Data availability
The current manuscript is a theoretical study, so no data have been generated for this manuscript. Code for computational analysis is publicly available via GitHub and listed in the Key Resources Table. We use data from *Dannenberg et al., 2020*.

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
