## [Editor Report · eLife Assessment]

This **valuable** study presents a theoretical framework in which spatial periodicity in grid cell firing emerges as the optimal solution for encoding two-dimensional spatial trajectories via sequential neural activation. The idea is supported by **solid** evidence, though it rests on several key assumptions that merit careful consideration. This work will be of interest to neuroscientists investigating the neural mechanisms underlying spatial navigation.

---

## [Referee Report · Reviewer #1 (Public review)]

Summary:

This manuscript aims to explain the emergence of grid-like spatial firing patterns. Rather than taking the existence of grid cells as a given and asking what does their properties say about their function, the authors reverse the approach: they begin with a proposed computational function that the brain may need to perform-coding of 2D spatial trajectories using sequences of neural activity-and ask what type of neural code would optimally support this function. They show that, under a set of formal assumptions, such a code leads to the emergence of spatial periodicity and a hexagonal grid pattern. The aim is to provide a normative explanation for the existence of grid cells grounded in functional constraints.

Strengths:

The manuscript presents a mathematically well-defined framework that is internally consistent. The derivation is structured and leads to a hexagonal lattice as the most efficient solution for representing directional trajectories. The authors provide comparisons to experimental observations and extend the model to explain several findings in the grid cell literature. In the revised version, the discussion of foundational assumptions is expanded, and the manuscript better situates itself in relation to prior theoretical work. Overall, this work adds a very interesting view to the broader conversation about the role and origin of grid cells by offering a theoretical alternative grounded in trajectory coding.

Weaknesses:

The model depends on assumptions that, while plausible, should be treated as chosen assumptions. These include the premise that (1) grid function is trajectory coding, (2) that trajectory coding is implemented through sequences of neural activity, and (3) that such sequences are largely independent of spatial position. In the revised manuscript, the authors provide more literature to support these assumptions.

---

## [Referee Report · Reviewer #2 (Public review)]

Summary:

In this work, the authors consider the required functional properties of neurons that trajectories in 2D space using cell sequences, ultimately linking the required properties to those found in grid cells. In their argument, the authors first introduce a set of definitions and axioms, which then lead to their conclusion that a hexagonal pattern is the most efficient or parsimonious pattern one could use to uniquely label different 2D trajectories using sequences of cells. The authors then go through a set of classic experimental results in the grid cell literature - e.g. that the grid modules exhibit a multiplicative scaling, that the grid pattern expands with novelty or is warped by reward, etc. - and describe how these results are either consistent with or predicted by their theory. Overall, this paper asks a very interesting question and provides an intriguing answer.

Major strengths:

The general idea behind the paper is very interesting - why *does* the grid pattern take the form of a hexagonal grid? This is a question that has been raised many times; finding a truly satisfying answer is difficult but of great interest. The authors' main assertion that the answer to this question has to do with the ability of a hexagonal arrangement of neurons to uniquely encode 2D trajectories is an intriguing suggestion. It is also impressive that the authors considered such a wide range of experimental results in relation to their theory.

Major weaknesses:

One weakness I perceive is that the paper overstates what it delivers. In the introduction, the authors claim to provide "mathematical proof that ... the nature of the problem being solved by grid cells is coding of trajectories in 2-D space using cell sequences. By doing so, we offer a specific answer to the question of why grid cell firing patterns are observed in the mammalian brain." By virtue of the fact that the authors make assumptions about biological function in their claims, this paper does not provide proof of what grid cells are doing to support behavior nor provide the true answer as to why grid patterns are found in the brain. Although I find this study both intriguing and important - and I respect the authors' perspective - as an experimentalist guided by the principle that biological theories are never proven but instead continually supported by data, suggestions of a proof of grid cell function are hard for me to get behind. Regardless, the paper presents a compelling line of reasoning that enhance our understanding of grid cells.

---

## [Referee Report · Reviewer #3 (Public review)]

Concerning the revised manuscript, the authors are to be commended for carefully addressing the reviewers' comments and updating the references cited.

I will differ with the authors' argument that Gardner et al's paper supports the idea of sequences, except in the most trivial sense, namely that topology implies continuity, and hence movement along the manifold will be continuous, and if one discretizes this movement, one would get a sequence.

This has very little to do with the idea that the authors propose in their manuscript. If the authors were to so choose, their idea will produce an embedded graph, and they could study the topology of this construct---but this would mean going off on a tangent.

Let us not hold up the authors to an impossible standard, namely that their theory should explain everything in the grid cell field. Not every finding under the sun needs be addressed in the discussion.

My own take on the manuscript had been that the authors' interesting idea might be fairly straightforwardly provable. The authors have decided to put another student on this particular project. That is perfectly OK. Just one final note: mathematically, the main focus ought to be on sequences of prime length (many other results will likely follow).

---

## [Author Response]

The following is the authors’ response to the original reviews

**eLife assessment**
This valuable study aims to present a mathematical theory for why the periodicity of the hexagonal pattern of grid cell firing would be helpful for encoding 2D spatial trajectories. The idea is supported by solid evidence, but some of the comparisons of theory to the experimental data seem incomplete, and the reasoning supporting some of the assumptions made should be strengthened. The work would be of interest to neuroscientists studying neural mechanisms of spatial navigation.

We thank the reviewers for this assessment. We have addressed the comments made by reviewers and believe that the revised manuscript has theoretical and practical implications beyond the subfield of neuroscience concerned with mechanisms underpinning spatial memory and spatial navigation. Specifically, the demonstration that four simple axioms beget the spatial firing pattern of grid cells is highly relevant for the field of artificial intelligence and neuromorphic computing. This relevance stems from the fact that the four axioms define a set of four simple computational algorithms that can be implemented in future work in grid cell-inspired computational algorithms. Such algorithms will be impactful because they can perform path integration, a function that is independent of an animal’s or agent’s location and therefore generalizable. Moreover, because of the functional organization of grid cells into modules, the algorithm is also scalable. Generalizability and scalability are two highly sought-after properties of brain-inspired computational frameworks. We also believe that the question why grid cells emerge in the brain is a fundamental one. This manuscript is, to our knowledge, the first one that provides an interpretable and intuitive answer to why grid cells are observed in the brain.

Before addressing each comment, we would like to point out that the first sentence of the assessment appears misphrased. The study does not aim to present a theory for why the periodicity in grid cell firing would be helpful for encoding 2D spatial trajectories. To present a theory “for why grid cell firing would be helpful for encoding 2D trajectories”, one assumes the existence of grid cells a priori. Instead of assuming the existence of grid cells and deriving a computational function from grid cells, our study derives grid cells from a computational function, as correctly summarized by reviewers #1 and #3 in their individual statements. In contrast to previous normative models, we prove mathematically that spatial periodicity in grid cell firing is implied by a sequence code of trajectories. If the brain uses cell sequences to code for trajectories, spatially periodic firing must emerge. As correctly pointed out by reviewer #1, the underlying assumptions of this study are that the brain codes for trajectories and that it does so using cell sequences. In response to comments by reviewer #1, we now discuss these two assumptions more rigorously.

**Public Reviews:**

**Reviewer #1 (Public Review):**
Rebecca R.G. et al. set to determine the function of grid cells. They present an interesting case claiming that the spatial periodicity seen in the grid pattern provides a parsimonious solution to the task of coding 2D trajectories using sequential cell activation. Thus, this work defines a probable function grid cells may serve (here, the function is coding 2D trajectories), and proves that the grid pattern is a solution to that function. This approach is somewhat reminiscent in concept to previous works that defined a probable function of grid cells (e.g., path integration) and constructed normative models for that function that yield a grid pattern. However, the model presented here gives clear geometric reasoning to its case.Stemming from 4 axioms, the authors present a concise demonstration of the mathematical reasoning underlying their case. The argument is interesting and the reasoning is valid, and this work is a valuable addition to the ongoing body of work discussing the function of grid cells.However, the case uses several assumptions that need to be clearly stated as assumptions, clarified, and elaborated on: Most importantly, the choice of grid function is grounded in two assumptions:(1) that the grid function relies on the activation of cell sequences, and(2) that the grid function is related to the coding of trajectories. While these are interesting and valid suggestions, since they are used as the basis of the argument, the current justification could be strengthened (references 28-30 deal with the hippocampus, reference 31 is interesting but cannot hold the whole case).

We thank this reviewer for the overall positive and constructive criticism. We agree with this reviewer that our study rests on two premises, namely that (1) a code for trajectories exist, and (2) this code is implemented by cell sequences. We now discuss and elaborate on the data in the literature supporting the two premises.

In addition to the work by Zutshi et al. (reference 31 in the original manuscript), we have now cited additional work presenting experimental evidence for sequential activity of neurons in the medial entorhinal cortex, including sequential activity of grid cells.

We have added the following paragraph to the Discussion section:

“Recent studies provided compelling evidence for sequential activity of neurons representing spatial trajectories. In particular, Gardner et al. (2022) demonstrated that the sequential activity of hundreds of simultaneously recorded grid cells in freely foraging rats represented spatial trajectories. Complementary preliminary results indicate that grid cells exhibit left-rightalternating “theta sweeps,” characterized by temporally compressed sequences of spiking activity that encode outwardly oriented trajectories from the current location (Vollan et al., 2024).

The concept of sequential grid cell activity extends beyond spatial coding. In various experimental contexts, grid cells have been shown to encode non-spatial variables. For instance, in a stationary auditory task, grid cells fired at specific sounds along a continuous frequency axis (Aronov et al., 2017). Further studies revealed that grid cell sequences also represent elapsed time and distance traversed, such as during a delay period in a spatial alternation task (Kraus et al., 2015). Similar findings were reported for elapsed time encoded by grid cell sequences in mice performing a virtual “Door Stop” task (Heys and Dombeck, 2018).

Additionally, spatial trajectories represented by temporally compressed grid cell sequences have been observed during sleep as replay events (Ólafsdóttir et al., 2016; O’Neill et al., 2017). Collectively, these studies demonstrate that sequential activity of neurons within the MEC, particularly grid cells, consistently encodes ordered experiences, suggesting a fundamental role for temporal structure in neuronal representations.

The theoretical underpinnings of grid cell activity coding for ordered experiences have been explored previously by Rueckemann et al. (2021) who argued that the temporal order in grid cell activation allows for the construction of topologically meaningful representations, or neural codes, grounded in the sequential experience of events or spatial locations. However, while Rueckemann et al. argue that the MEC supports temporally ordered representations through grid cell activity, our findings suggest an inverse relationship: namely, that grid cell activity emerges from temporally ordered spatial experiences. Additional studies demonstrate that hippocampal place cells may derive their spatial coding properties from higher-order sequence learning that integrates sensory and motor inputs (Raju et al., 2024) and that hexagonal grids, if assumed a priori, optimally encode transitions in spatiotemporal sequences (Waniek, 2018).

Together, experimental and theoretical evidence demonstrate the significance of sequential neuronal activity within the hippocampus and entorhinal cortex as a core mechanism for representing both spatial and temporal information and experiences.”

The work further leans on the assumption that sequences in the same direction should be similar regardless of their position in space, it is not clear why that should necessarily be the case, and how the position is extracted for similar sequences in different positions.

We thank this reviewer for giving us the opportunity to clarify this point. We define a trajectory as a path taken in space (Definition 6). By this definition, a code for trajectories is independent of the animal’s spatial location. This is consistent with the definition of path integration, which is also independent of an animal’s spatial location. If the number of neurons is finite (Axiom #4) and the space is large, sequences must eventually repeat in different locations. This results in neural sequences coding for the same directions being identical at different locations. We have clarified this point under new Remark 6.1. in the Results section of the revised:

“Remark 6.1. Note that a code for trajectories is independent of the animal’s spatial location, consistent with the definition of path integration. This implies that, if the number of neurons is finite (Axiom #4) and the space is large, sequences must eventually repeat in different location, resulting in neural sequences coding for the same trajectories at different locations.”

The formal proof was already included in the original manuscript: “Generally speaking, starting in a firing field of element i and going along any set of firing fields, some element must eventually become active again since the total number of elements is finite by axiom 4. Once there is a repeat of one element’s firing field, the whole sequence of firing fields of all elements must repeat by axiom 1. More specifically, if we had a sequence 1,2, … , k, 1, t of elements, then 1,2 and 1, t both would code for traveling in the same direction from element 1, contradicting axiom 1.”

Further: “More explicitly, assuming axioms 1 and 4, the firing fields of trajectory-coding elements must be spatially periodic, in the sense that starting at any point and continuing in a single direction, the initial sequence of locally active elements must eventually repeat with a repeat length of at least 3”.

Regarding the question how an animal’s position is extracted for similar sequences in different positions, we agree with this reviewer that this is an important question when investigating the contributions of grid cells to the coding of space. However, since a code for trajectories is independent of spatial location, the question of how to extract an animal’s position from a trajectory code is irrelevant for this study.

While a trajectory code by neural sequences begets grid cells, a spatial code by neural sequences does not. Nevertheless, grid cells could contribute to the coding of space (in addition to providing a trajectory code). However, while experimental evidence from studies with rodents and human subjects and theoretical work demonstrated the importance of grid cells for path integration (Fuhs and Touretzky, 2006; McNaughton et al., 2006; Moser et al., 2017), experimental studies have shown that grid cells contribute little to the coding of space by place cells (Hales et al., 2014). Yet, theoretical work (Mathis et al., 2012) showed that coherent activity of grid cells across different modules can provide a code for spatial location that is more accurate than spatial coding by place cells in the hippocampus. Importantly, such a spatial code by coherent activity across grid cell modules does not require location-dependent differences in neural sequences.

The authors also strengthen their model with the requirement that grid cells should code for infinite space. However, the grid pattern anchors to borders and might be used to code navigated areas locally. Finally, referencing ref. 14, the authors claim that no existing theory for the emergence of grid cell firing that unifies the experimental observations on periodic firing patterns and their distortions under a single framework. However, that same reference presents exactly that - a mathematical model of pairwise interactions that unifies experimental observations. The authors should clarify this point.

We thank this reviewer for this valuable feedback. We agree that grid cells anchor to borders and may be used to code navigated areas locally. In fact, the trajectory code performs a local function, namely path integration, and the global grid pattern can only emerge from performing this local computation if the activity of at least one grid unit or element (we changed the wording from unit to element based on feedback from reviewer #3) is anchored to either a spatial location or a border. Yet, the trajectory code itself does not require anchoring to a reference frame to perform local path integration. Because of the local nature of the trajectory code, path integration can be performed locally without the emergence of a global grid pattern. This has been shown experimentally in mice performing a path integration task where changes in the location of a task-relevant object resulted in translations of grid patterns in single trials. Although no global grid pattern was observed, grid cells performed path integration locally within the multiple reference frames defined by the task-relevant object, and grid patterns were visible when the changes in the references frames were accounted for in computing the rate maps (Peng et al., 2023). The data by Peng et al. (2023) confirm that the anchoring of the grid pattern to borders and the emergence of the global pattern are not required for local coding of trajectories. The global pattern emerges only when the reference frame does not change. However, this global pattern itself might not serve any function. According to the trajectory code model, the beguiling grid pattern is merely a byproduct of a local path integration function that is independent of the animal’s current location (which makes the code generalizable across space). The reviewer is correct that, if the reference frame used to anchor the grid pattern did not change in infinite space, the trajectory code model of grid cell firing would predict an infinite global pattern. But does the proof implicitly assume that space is infinite? The trajectory code model makes the quantitative prediction that the field size increases linearly with an increase in grid spacing (the distance between two fields). If the field size remains fixed, periodicity will emerge in finite spaces that are larger than the grid spacing. We have clarified these points in the revised manuscript:

“Notably, the trajectory code itself does not require anchoring to a reference frame to perform local path integration. Because of the local nature of the trajectory code, path integration can be performed locally without the emergence of a global grid pattern. This has been shown experimentally in mice performing a path integration task where changes in the location of a task-relevant object resulted in translations of grid patterns in single trials (Peng et al., 2023). Although no global grid pattern was observed because the reference frame was not fixed in space, grid cells performed path integration locally within the reference frame defined by the moving task-relevant object, and grid patterns were visible when the changes in the references frames were accounted for in computing the rate maps”.

Regarding how the emergence of grid cells from a trajectory code relates to the theory of a local code by grid cells brought forward by Ginosar et al. (ref. 14), we argue that the local computational function suggested by Ginosar et al. is to provide a code for trajectories. The perspective article by Ginosar et al. provides an excellent review of the experimental data on grid cells that point to grid cells performing a local function (see also Kate Jeffery’s excellent review article (Jeffery, 2024) on the mosaic structure of the mammalian cognitive map.) Assuming the existence of grid cells a priori, Ginosar et al. then propose three possible functions of grid cells, all of which are consistent with the trajectory code model of grid cell firing. Yet, the perspective article remains agnostic, in our opinion, on the exact nature of the local computation that is carried out by grid cells. But without knowing the local computation underlying grid cell function, a unifying theory explaining the emergence of grid cells cannot be considered complete. In contrast, our manuscript identifies the local computational function as a trajectory code by cell sequences. We have clarified these points in the revised manuscript:

“The influential hypothesis that grid cells provide a universal map for space is challenged by experimental data suggesting a yet to be identified local computational function of grid cells (Ginosar et al., 2023; Jeffery, 2024). Here, we identify this local computational function as a trajectory code.”

The mathematical model of pairwise interactions described by Ginosar et al. is fundamentally different from the mathematical framework developed in our manuscript. The mathematical model by Ginosar et al. describes how pairwise interactions between already existent grid fields can explain distortions in the grid pattern caused by the environment’s geometry, reward zones, and dimensionality. However, the model does not explain why there is a grid pattern in the first place. In contrast, our trajectory model provides an explanation for why grid cells may exist by demonstrating that a grid pattern emerges from a trajectory code by cell sequences. We stand by our assessment that a unifying theory of grid cells is not complete if it takes the existence of the grid pattern for granted.

**Reviewer #2 (Public Review):**
Summary:In this work, the authors consider why grid cells might exhibit hexagonal symmetry - i.e., for what behavioral function might this hexagonal pattern be uniquely suited? The authors propose that this function is the encoding of spatial trajectories in 2D space. To support their argument, the authors first introduce a set of definitions and axioms, which then lead to their conclusion that a hexagonal pattern is the most efficient or parsimonious pattern one could use to uniquely label different 2D trajectories using sequences of cells. The authors then go through a set of classic experimental results in the grid cell literature - e.g. that the grid modules exhibit a multiplicative scaling, that the grid pattern expands with novelty or is warped by reward, etc. - and describe how these results are either consistent with or predicted by their theory. Overall, this paper asks a very interesting question and provides an intriguing answer. However, the theory appears to be extremely flexible and very similar to ideas that have been previously proposed regarding grid cell function.

We thank this reviewer for carefully reading the manuscript and their valuable feedback which helps us clarify major points of the study. One major clarification is that the theoretical/axiomatic framework we put forward does not assume grid cells a priori. In contrast, we start by hypothesizing a computational function that a brain region shown to be important for path integration likely needs to solve, namely coding for spatial trajectories. We go on to show that this computational function begets spatially periodic firing (grid maps). By doing so, we provide mathematical proof that grid maps emerge from solving a local computational function, namely spatial coding of trajectories. Showing the emergence of grid maps from solving a local computational function is fundamentally different from many previous studies on grid cell function, which assign potential functions to the existing grid pattern. As we discuss in the manuscript, our work is similar to using normative models of grid cell function. However, in contrast to normative models, we provide a rigorous and interpretable mathematical framework which provides geometric reasoning to its case.

Major strengths:The general idea behind the paper is very interesting - why *does* the grid pattern take the form of a hexagonal grid? This is a question that has been raised many times; finding a truly satisfying answer is difficult but of great interest to many in the field. The authors' main assertion that the answer to this question has to do with the ability of a hexagonal arrangement of neurons to uniquely encode 2D trajectories is an intriguing suggestion. It is also impressive that the authors considered such a wide range of experimental results in relation to their theory.

We thank this reviewer for pointing out the significance of the question addressed by our manuscript.

Major weaknesses:One major weakness I perceive is that the paper overstates what it delivers, to an extent that I think it can be a bit confusing to determine what the contributions of the paper are. In the introduction, the authors claim to provide "mathematical proof that ... the nature of the problem being solved by grid cells is coding of trajectories in 2-D space using cell sequences. By doing so, we offer a specific answer to the question of why grid cell firing patterns are observed in the mammalian brain." This paper does not provide proof of what grid cells are doing to support behavior or provide the true answer as to why grid patterns are found in the brain. The authors offer some intriguing suggestions or proposals as to why this might be based on what hexagonal patterns could be good for, but I believe that the language should be clarified to be more in line with what the authors present and what the strength of their evidence is.

We thank this reviewer for this assessment. While there is ample experimental evidence demonstrating the importance of grid cells for path integration, we agree with this reviewer that there may be other computational functions that may require or largely benefit from the existence of grid cells. We now acknowledge the fact that we have provided a likely teleological cause for the emergence of grid cells and that there might be other causes for the emergence of grid cells. We have changed the wording in the abstract and discussion sections to acknowledge that our study does provide a likely teleological cause. We choose “likely” because the computational function – trajectory coding – from which grid maps emerge is very closely associated to path integration, which numerous experimental and theoretical studies associate with grid cell function.

Relatedly, the authors claim that they find a teleological reason for the existence of grid cells - that is, discover the function that they are used for. However, in the paper, they seem to instead assume a function based on what is known and generally predicted for grid cells (encode position), and then show that for this specific function, grid cells have several attractive properties.

We agree with this reviewer that we leveraged what is known about grid cells, in particular their importance for path integration, in finding a likely teleological cause. However, the major significance of our work is that we demonstrate that coding for spatial trajectories requires spatially periodic firing (grid cells).This is very different from assuming the existence of grid cells a priori and then showing that grid cells have attractive, if not optimal, properties for this function. If we had shown that grid cells optimized a code for trajectories, this reviewer would be correct: we would have suggested just another potential function of grid cells. Instead, we provide both proof and intuition that trajectory coding by cell sequences begets grid cells (not the other way around), thereby providing a likely teleological cause for the emergence of grid cells. As stated above, we clarified in the revised manuscript that we provide a likely teleological cause which requires additional experimental verification.

There is also some other work that seems very relevant, as it discusses specific computational advantages of a grid cell code but was not cited here: https://www.nature.com/articles/nn.2901.

We thank this reviewer for pointing us toward this article by (Sreenivasan and Fiete, 2011). The revised manuscript now cites this article in the Introduction and Discussion sections. We agree that the article by (Sreenivasan and Fiete, 2011) discusses a specific computational advantage of a population code by grid cells, namely unprecedented robustness to noise in estimating the location from the spiking information of noisy neurons. However, the work by (Sreenivasan and Fiete, 2011) differs from our work in that the authors assume the existence of grid cells a priori.

In addition, we now discuss other relevant work, namely work on the conformal isometry hypothesis by (Schøyen et al., 2024) and (Xu et al., 2024), published as pre-prints after publication of the first version of our manuscript, as well as work on transition scale- spaces by Nicolai Waniek. (Xu et al., 2024) and (Schøyen et al., 2024) investigate conformal isometry in the coding of space by grid cells. Conformal isometry means that trajectories in neural space map trajectories in physical space. (Xu et al., 2024) show that the conformal isometry hypothesis can explain the spatially periodic firing pattern of grid cells. (Schøyen et al., 2024) further show that a module of seven grid cells emerges if space is encoded as a conformal isometry, ensuring equal representation in all directions. While the work by (Xu et al., 2024) and (Schøyen et al., 2024) arrive at very similar conclusions as stated in the current manuscript, the conformal isometry hypothesis provides only a partial answer to why grid cells exist because it doesn’t explain why conformal isometry is important or required. In contrast, a sequence code of trajectories provides an intuitive answer to why such a code is important for animal behavior. Furthermore, we included the work by Nicolai Waniek, (2018, 2020) in the Discussion, who demonstrated that the hexagonal arrangement of grid fields is optimal for coding transitions in space.

The paragraph added to the Discussion reads as follows:

“As part of the proof that a trajectory code by cell sequences begets spatially periodic firing fields, we proved that the centers of the firing fields must be arranged in a hexagonal lattice. This arrangement implies that the neural space is a conformally isometric embedding of physical space, so that local displacements in neural space are proportional to local displacements of an animal or agent in physical space, as illustrated in Figure 5. This property has recently been introduced in the grid cell literature as the conformal isometry hypothesis(Schøyen et al., 2024; Xu et al., 2024). Strikingly, Schøyen et al.(Schøyen et al., 2024) arrive at similar if not identical conclusions regarding the geometric principles in the neural representations of space by grid cells.”

A second major weakness was that some of the claims in the section in which they compared their theory to data seemed either confusing or a bit weak. I am not a mathematician, so I was not able to follow all of the logic of the various axioms, remarks, or definitions to understand how the authors got to their final conclusion, so perhaps that is part of the problem. But below I list some specific examples where I could not follow why their theory predicted the experimental result, or how their theory ultimately operated any differently from the conventional understanding of grid cell coding. In some cases, it also seemed that the general idea was so flexible that it perhaps didn't hold much predictive power, as extra details seemed to be added as necessary to make the theory fit with the data.I don't quite follow how, for at least some of their model predictions, the 'sequence code of trajectories' theory differs from the general attractor network theory. It seems from the introduction that these theories are meant to serve different purposes, but the section of the paper in which the authors claim that various experimental results are predicted by their theory makes this comparison difficult for me to understand. For example, in the section describing the effect of environmental manipulations in a familiar environment, the authors state that the experimental results make sense if one assumes that sequences are anchored to landmarks. But this sounds just like the classic attractornetwork interpretation of grid cell activity - that it's a spatial metric that becomes anchored to landmarks.

We thank this reviewer for giving us the opportunity to clarify in what aspects the ‘sequence code of trajectories’ theory of grid cell firing differs from the classic attractor network models, in particular the continuous attractor network (CAN) model. First of all, the CAN model is a mechanistic model of grid cell firing that is specifically designed to simulate spatially periodic firing of grid cells in response to velocity inputs. In contrast, the sequence code of trajectories theory of grid cell firing resembles a normative model showing that grid cells emerge from performing a specific function. However, in contrast to previous normative models, the sequence code of trajectories model grounds the emergence of grid cell firing in a mathematical proof and both geometric reasoning and intuition. The proof demonstrates that the emergence of grid cells is the only solution to coding for trajectories using cell sequences. The sequence code of trajectories model of grid cell firing is agnostic about the neural mechanisms that implements the sequence code in a population of neurons. One plausible implementation of the sequence code of trajectories is in fact a CAN. In fact, the sequence code of trajectories theory predicts conformal isometry in the CAN, i.e., a trajectory in neural space is proportional to a trajectory of an animal in physical space. However, other mechanistic implementations are possible. We have clarified how the sequence code of trajectories theory of grid cells relates to the mechanistic CAN models of grid cells.

We added the following text to the Discussion section:

“While the sequence code of trajectories-model of grid cell firing is agnostic about the neural mechanisms that implements the sequence code, one plausible implementation is a continuous attractor network (McNaughton et al., 2006; Burak and Fiete, 2009). Interestingly, a sequence code of trajectories begets conformal isometry in the attractor network, i.e., a trajectory in neural space is proportional to a trajectory of an animal in physical space.”

It was not clear to me why their theory predicted the field size/spacing ratio or the orientation of the grid pattern to the wall.

We thank this reviewer for bringing to our attention that we lacked a proper explanation for why the sequence code of trajectories theory predicts the field size/spacing ration in grid maps. We have modified/added the following text to the Results section of the manuscript to clarify this point:

“Because the sequence code of trajectories model of grid cell firing implies a dense packing of firing fields, the spacing between two adjacent grid fields must change linearly with a change in field size. It follows that the ratio between grid spacing and field size is fixed. When using the distance between the centers of two adjacent grid fields to measure grid spacing and a diameter-like metric to measure grid field size, we can compute the ratio of grid spacing to grid field size as √7≈2.65 (see Methods).”

We are also grateful for this reviewer’s correctly pointing out that the explanation as to why the sequence code of trajectories predicts a rotation of the grid pattern relative to a set of parallel walls in a rectangular environment. We have now made explicit the underlying premise that a sequence of firing fields from multiple grid cells are aligned in parallel to a nearby wall of the environment. We cite additional experimental evidence supporting this premise. Concretely, we quote Stensola and Moser summarizing results reported in (Stensola et al. 2015): “A surprising observation, however, was that modules typically assumed one of only four distinct orientation configurations relative to the environment” (Stensola and Moser, 2016). Importantly, all of the four distinct orientations show the characteristic angular rotation. Intriguingly, this is predicted by the sequence code of trajectories-model under the premise that a sequence of firing fields aligns with one of the geometric boundaries of the environment, as shown in Author response image 1 below.

**Author response image 1. sa4fig1:** Under the premise that a sequence of firing fields aligns with one of the geometric boundaries (walls) of a square arena, there are precisely four possible distinct configurations of orientations. This is precisely what has been observed in experiments (Stensola et al., 2015; Stensola and Moser, 2016).

We added clarifying language to the Results section: “Under the premise that a sequence of firing fields aligns with one of the geometric boundaries of the environment, the sequence code model explains that the grid pattern typically assume one of only four distinct orientation configurations relative to the environment41,46. Concretely, the four orientation configurations arise when one row of grid fields aligns with one of the two sets of parallel walls in a rectangular environment, and each arrangement can result in two distinct orientations (Figure 3B).”

I don't understand how repeated advancement of one unit to the next, as shown in Figure 4E, would cause the change in grid spacing near a reward.

In familiar environments, spatial firing fields of place cells in hippocampal CA1 and CA3 tend to shift backwards with experience (Mehta et al., 2000; Lee et al., 2004; Roth et al., 2012; Geiller et al., 2017; Dong et al., 2021). This implies that the center of place fields move closer to each other. A potential mechanism has been suggested, namely NMDA receptor-dependent longterm synaptic plasticity (Ekstrom et al., 2001). When we apply the same principle observed for place fields on a linear track to grid fields anchored to a reward zone, grid fields will “gravitate” towards the reward side. A similar idea has been presented by (Ginosar et al., 2023) who use the analogy of reward locations as “black holes”. In contrast to (Ginosar et al., 2023), who we cite multiple times, our idea unifies observations on place cells and grid cells in 1-D and 2-D environments and suggests a potential mechanism. We changed the wording in the revised manuscript and clarified the underlying premises.

I don't follow how this theory predicts the finding that the grid pattern expands with novelty. The authors propose that this occurs because the animals are not paying attention to fine spatial details, and thus only need a low-resolution spatial map that eventually turns into a higher-resolution one. But it's not clear to me why one needs to invoke the sequence coding hypothesis to make this point.

We agree with this reviewer that this point needs clarification. The sequence code model adds explanatory power to the hypothesis that the grid pattern in a novel environment reflects a lowresolution mapping of space or spatial trajectories because it directly links spatial resolution to both field size and spacing of a grid map. Concretely, the spatial resolution of the trajectory code is equivalent to the spacing between two adjacent spatial fields, and the spatial resolution is directly proportional to the grid spacing and field size. If one did not evoke the sequence coding hypothesis, one would need to explain how and why both spacing and field size are related to the spatial resolution of the grid map. Lastly, as written in the manuscript text, we point out that, while the experimentally observed expansion of grid maps is consistent with the sequence code of trajectory, it is not predicted by the theory without making further assumption.

The last section, which describes that the grid spacing of different modules is scaled by the square root of 2, says that this is predicted if the resolution is doubled or halved. I am not sure if this is specifically a prediction of the sequence coding theory the authors put forth though since it's unclear why the resolution should be doubled or halved across modules (as opposed to changed by another factor).

We agree with reviewer #2 that the exact value of the scaling factor is not predicted by the sequence coding theory. E.g., the sequence code theory does not explain why the spatial resolution doesn’t change by a factor 3 or 1.5 (resulting in changes in grid spacing by square root of 3 or square root of 1.5, respectively). We have changed the wording to reflect this important point. We further clarified in the revised manuscript that future work on multiscale representations using modules of grid cells needs to show why changing the spatial resolution across modules by a factor of 2 is optimal. Interestingly, a scale ratio of 2 is commonly used in computer vision, specifically in the context of mipmapping and Gaussian pyramids, to render images across different scales. Literature in the computer vision field describes why a scaling factor of 2 and the use of Gaussian filter kernels (compare with Gaussian firing fields) is useful in allowing a smooth and balanced transition between successive levels of an image pyramid (Burt and Adelson, 1983; Lindeberg, 2008). Briefly, larger factors (like 3) could result in excessive loss of detail between levels, while smaller factors (like 1.5) would not reduce the image size enough to justify additional levels of computation (that would come with the structural cost of having more grid cell modules in the brain). We have clarified these points in the Discussion section.

**Reviewer #3 (Public Review):**
The manuscript presents an intriguing explanation for why grid cell firing fields do not lie on a lattice whose axes aligned to the walls of a square arena. This observation, by itself, merits the manuscript's dissemination to the eLife's audience.

We thank this reviewer for their positive assessment.

The presentation is quirky (but keep the quirkiness!).

We kept the quirkiness.

But let me recast the problem presented by the authors as one of combinatorics. Given repeating, spatially separated firing fields across cells, one obtains temporal sequences of grid cells firing. Label these cells by integers from [n]. Any two cells firing in succession should uniquely identify one of six directions (from the hexagonal lattice) in which the agent is currently moving.Now, take the symmetric group Σ of cyclic permutations on n elements. We ask whether there are cyclic permutations of [n] such that(πi+1−πi)modn≠±1modn,∀i.So, for instance, (4,2,3,1) would not be counted as a valid permutation of (1,2,3,4), as (2,3) and (1,4) are adjacent.Furthermore, given [n], are there two distinct cyclic permutations such that *no* adjacencies are preserved when considering any pair of permutations (among the triple of the original ordered sequence and the two permutations)? In other words, if we consider the permutation required to take the first permutation into the second, that permutation should not preserve any adjacencies.**Key question**: is there any difference between the solution to the combinatorics problem sketched above and the result in the manuscript? Specifically, the text argues that for n=7 there is only *one* solution.Ideally, one would strive to obtain a closed-form solution for the number of such permutations as a function of n.

This is a great question! We currently have a student working on describing all possible arrangements of firing fields (essentially labelings of the hexagonal lattice) that satisfy the axioms in 2D, and we expect that results on the number of such arrangements will come out of his work. We plan to publish those results separately, possibly targeting a more mathematical audience.

The argument above appears to only apply in the case that every row (and every diagonal) contains all of the elements 1,...,n. However, when n is not prime, there are often arrangements where rows and/or diagonals do not contain every element from 1,...,n. For example, some admissible patterns with 9 neurons have a repeat length of 3 in all directions (horizontally and both diagonals). As a result the construction listed here will not give a full count of all possible arrangements.

**Recommendations for the authors:**

**Reviewer #1 (Recommendations For The Authors):**
I think the concise style of mathematical proof is both a curse and a blessing. While it delivers the message, I think the fluency and readability of the mathematical proof could be improved with longer paragraphs and some more editing.

We have added some clarifications in the text that we hope improve the readability.

**Reviewer #3 (Recommendations For The Authors):**
A minor qualm I have with the nomenclature:On page 7:“To prove this statement, suppose that row A consists of units 1,…,k repeating in this order. Then any row that contains any unit from 1,…,k must contain the full repeat 1,…,k by axiom 1. So any row containing any unit from 1,…,k is a translation of row A, and any unit that does not contain them is disjoint from row A.”The last use of `unit' at the end of this paragraph instead of `row' is confusing. Technically, the authors have given themselves license to use this term by defining a unit to be “either to a single cell or a cell assembly”. Yet modern algebra tends to use `unit' as meaning a ring element that has an inverse.

We have renamed “unit” to “element” to avoid confusion with the terminology in modern algebra.